# Small Intestine Tumors: Diagnostic Role of Multiparametric Ultrasound

**DOI:** 10.3390/healthcare13212776

**Published:** 2025-10-31

**Authors:** Kathleen Möller, Christian Jenssen, Klaus Dirks, Alois Hollerweger, Heike Gottschall, Siegbert Faiss, Christoph F. Dietrich

**Affiliations:** 1Medical Department I/Gastroenterology, Sana Hospital Lichtenberg, 10365 Berlin, Germany; k.moeller@live.de (K.M.); heike.gottschall@sana.de (H.G.); siegbert.faiss@sana.de (S.F.); 2Department of Internal Medicine, Krankenhaus Märkisch-Oderland, 15344 Strausberg, Germany; c.jenssen@khmol.de; 3Brandenburg Institute for Clinical Ultrasound, Medical University Brandenburg, 16816 Neuruppin, Germany; 4Medical Clinic II, University Hospital Würzburg, 97070 Würzburg, Germany; klaus.dirks@t-online.de; 5Department of Radiology, Hospital Barmherzige Brüder, 5020 Salzburg, Austria; alois.hollerweger@aon.at; 6Ultrasound Center of Education and Excellence, 3626 Hünibach, Switzerland

**Keywords:** small intestine tumors, ultrasound, adenocarcinoma, neuroendocrine neoplasms, lymphoma, GIST

## Abstract

Small intestine tumors are rare. The four main groups include adenocarcinomas, neuroendocrine neoplasms (NEN), lymphomas, and mesenchymal tumors. The jejunum and ileum can only be examined endoscopically with device-assisted enteroscopy techniques (DAET), which are indicated only when specific clinical or imaging findings are present. The initial diagnosis of tumors of the small intestine is mostly made using computed tomography (CT). Video capsule endoscopy (VCE), computed tomography (CT) enterography, and magnetic resonance (MR) enterography are also time-consuming and costly modalities. Modern transabdominal gastrointestinal ultrasound (US) with high-resolution transducers is a dynamic examination method that is underrepresented in the diagnosis of small intestine tumors. US can visualize wall thickening, loss of wall stratification, luminal stenosis, and dilatation of proximal small-intestinal segments, as well as associated lymphadenopathy. This review aims to highlight the role and imaging features of ultrasound in the diagnosis of small-intestinal tumors.

## 1. Introduction

Small intestine tumors are rare worldwide. There are four main groups: adenocarcinomas, neuroendocrine neoplasms, lymphomas and mesenchymal tumors. Tumors in the small intestine are classified according to the WHO 2019 classification [1,2,3,4].

Although the small intestine has a length of 5–6 m, the global incidence of small intestine tumors is generally less than 1.0 per 100,000 and ranges from 0.3 to 2.0 per 100,000 inhabitants [5]. The most common types were neuroendocrine neoplasms (NEN) (33.2%), followed by adenocarcinomas (30.1%), lymphomas (16.2%), and GIST (7.1%). The remaining 13.5% were a collection of various other entities. The most common tumor in the duodenum was adenocarcinoma (52.8%). Adenocarcinoma was also the most common tumor in the jejunum (38.9%), followed by lymphomas (21.8%), NEN (12.4%), and GIST (9.3%). All other tumor entities accounted for 17.6%. NEN were most common in the ileum (62.8%), followed by lymphomas (15.9%), adenocarcinomas (14.7%), and GISTs (9.0%). The other various entities accounted for 9.6% [6].

The incidence of small intestine tumors and in particular of NEN has increased in recent decades [7]. In the United States, the incidence has more than doubled between 1975 (1.1/100,000) and 2018 (2.4/100,000) with a change in the leading entity from adenocarcinoma (1985: 42%; 2005: 33%) to NEN (1985: 28%; 2005: 44%). The proportion of mesenchymal tumors (in particular GIST; approximately 17%) and lymphomas (approximately 7%) remained stable [8]. More recent data show a continuation of these trends [9]. An increased risk of small intestine neoplasms has been observed in patients with Crohn’s disease, celiac disease, familial adenomatous polyposis and Peutz-Jeghers syndrome [1].

The standardized US-examination of the gastrointestinal tract (GIUS) is described in the EFSUMB guidelines and other publications [10,11,12,13]. EFSUMB GIUS guidelines describe the role of GIUS for chronic inflammatory bowel diseases [14], acute gastrointestinal diseases and emergencies [15,16], rare gastrointestinal diseases [17], and functional gastrointestinal disorders [18]. The US diagnosis of small intestinal tumors is described in the EFSUMB guideline on celiac disease and rare diseases [17,19]. Although US is a dynamic basic examination method with multiparametric procedures, the method is underrepresented in the diagnosis and characterization of small intestine tumors.

The following overview focuses on small intestine tumors, the possibilities for diagnosing them using US, and their specific appearance in US.

## 2. Search Strategy

We have based our classification of the various tumors in the small intestine on the 2019 WHO classification. The literature search and analysis were performed specifically for US characteristics of small intestine tumors. PubMed was the database in which entries were searched until 7 June 2025. The keywords and binary operators used are listed in brackets. A total of 3156 titles were reviewed. The abstracts were reviewed, and suitable articles were selected for further analysis. Similar articles on PubMed database were also checked for relevant abstracts: (small intestine adenocarcinoma OR small bowel tumor OR small intestine neuroendocrine tumor OR small intestine lymphoma OR small intestine mesenchymal tumor OR small intestine GIST OR Jejunum GIST OR small intestine leiomyoma OR small intestine Vanek’s tumor OR small intestine inflammatory myofibroblastic tumor OR small intestine lipoma OR secondary small intestine tumors OR small intestine melanoma) AND (Ultrasound OR Ultrasonography OR CEUS) (Figure 1). Further additions were made during the peer review process.

## 3. Symptoms—When Should Small Intestine Tumors Be Considered?

Tumors of the small intestine can be asymptomatic and an incidental finding. When they increase in size and obstruct the bowel lumen, symptoms occur ranging from nonspecific abdominal discomfort to cramp-like pain due to lumen stenosis and even ileus. Just like other gastrointestinal tract tumors, small intestine tumors can manifest with anemia, occult or overt gastrointestinal bleeding. There are few data available on the symptoms leading to diagnosis. In a retrospective analysis of 66 small intestine tumors, 28.8% of cases were discovered by chance. The most common symptom was abdominal pain (44.5%), while 12.1% of cases were presented with hematochezia and 10.6% with anemia [20]. Small bowel intussusception in adults should always be a reason to look for a small intestine tumor distal to the localization of intussusception. It is not uncommon that small intestine tumors cause nonspecific symptoms persisting for several months without diagnostic tests yielding any conclusive findings [21].

## 4. Imaging the Small Intestines

The role of standard endoscopic examinations is limited: The duodenum until the second portion is examined during every esophago-gastro-duodenoscopy, the terminal ileum during ileocolonoscopy, and the remnant duodenum and the upper jejunum can be viewed during push enteroscopy.

### 4.1. Device–Assisted Enteroscopy Techniques (DAET)

Device-assisted enteroscopy techniques (DAET), including double-balloon enteroscopy (DBE), single-balloon enteroscopy (SBE), and spiral overtube systems, are well-established methods for evaluating the small intestine [22,23,24]. These are the only endoscopic modalities which allow to examine the lumen and mucosa of the small intestine and to take forceps biopsies for histological evaluation of tumors. Depending on the suspected localization, DAET can be performed perorally or peranally.

In 2017, motorized spiral enteroscopy (MSE) was introduced, showing promise with shorter procedure times, deeper insertion, and higher complete enteroscopy rates than previous methods [25,26,27]. However, in July 2023, the device was withdrawn from the market due to safety concerns after a fatal case of esophageal perforation.

DAET requires laxative measures to prepare a clean intestinal lumen. Single-center studies and a few multicenter studies describe a high diagnostic yield of DAET for the detection and characterization of small intestine tumors [28,29,30,31]. However, the method is time-consuming, and inspection of the entire small intestine is not always possible. Because of its relative invasiveness, less invasive procedures are typically used as the first-line examination of the small intestine. DAET is indicated when findings in the small intestine have been diagnosed in other imaging or VCE examinations and it is advisable to perform a histological examination that is otherwise either not possible or for which primary surgery is not indicated, or for the endoscopic treatment of findings in the small intestine. For example, the treatment of lesions with gastrointestinal bleeding.

### 4.2. Video Capsule Endoscopy (VCE)

The search for an unclear source of bleeding is a main indication for video capsule endoscopy (VCE). The detection rate for small intestine tumors in VCE was 2.3% in a large study with 5129 patients [32]. In a meta-analysis with 106 small intestine tumors in 691 patients, 18.9% of tumors were missed in VCE [33]. It can be difficult to distinguish subepithelial tumors from extrinsic impressions. Residual staining and intense peristalsis with rapid passage in sections can also be limiting. If occult gastrointestinal bleeding cannot be adequately clarified by gastroscopy, push enteroscopy, ileocolonoscopy, or ultrasound, VCE is indicated. It is important that obstructions are ruled out beforehand, as otherwise capsule retention may occur, requiring surgical removal.

### 4.3. Endoscopic Ultrasound (EUS)

Mucosal and subepithelial intramural processes in the duodenum can be assessed using EUS. The jejunum and ileum cannot be reached using conventional EUS probes. It is possible to examine protruding impressions in the small intestine lumen using mini probes via DAET [34,35].

### 4.4. Abdominal CT, CT-/MR Enterography

Abdominal CT is often the first procedure or follows US in the evaluation of abdominal complaints. In a study by Rangiah et al. [36], CT showed a sensitivity of 87.7% for detecting small intestine tumors (specificity, PPV, and NPV) were not specified. In general, CT is a staging examination used to classify tumors before determining appropriate therapy. Furthermore, the possibility of a secondary tumor should always be considered, especially in older patients [37].

For CT or MR enterography, the lumen of the small intestine must be filled with a neutral contrast medium either by a duodenal catheter or by oral contrast uptake of approximately 1–2 L in 1.5–2 h. In addition to intraluminal contrast agents, intravenous iodinated contrast agents are also administered during CT. Gadolinium-based contrast agents are generally used in MRI for the assessment of inflammatory or potentially malignant small bowel processes. Study data seem to indicate that MRI has a higher diagnostic performance than CT in the specific subgroups of malignant small intestine tumors [38]. The overall sensitivity, specificity and accuracy in the diagnosis of small intestine lesions were 75.9%, 94.8% and 88.0% for CT enterography and 92.6%, 99.0% and 96.7% for MR enterography [38]. CT enterography is associated with radiation exposure. A standard CT enterography protocol (without dose reduction procedure) generally leads to an estimated effective radiation dose of 12 to 20 mSv. The use of low-dose techniques leads to an average dose reduction of 53 to 69% (from 12–20 to 5–7 mSv) for single-phase CT enterography examinations [39]. If unclear obstruction symptoms or occult gastrointestinal bleeding cannot be sufficiently clarified by gastroscopy, push enteroscopy, ileocolonoscopy, and US, MR or CT enterography is the next diagnostic step. MRI is preferable because there is no radiation exposure.

### 4.5. Transabdominal Ultrasound (US)

Few publications describe the role of US for the detection and characterization of small intestine tumors. Most cohort studies and registries rely on CT, MRI, or endoscopic findings for initial diagnosis and staging. Case series on small intestine tumors often do not include US, or only a minority of patients undergo US as part of the diagnostic workup. One main reason for underutilization of US is the imaging challenge arising from the fact that the small intestine has a length of 5–6 m and bowel loops lying folded on top of each other. Moreover, the small intestine is mobile, and the position of local findings can vary with intestinal peristalsis. Tumor detection may be limited by small size, deep location, overlying gas, and patient body habits. However, a standardized, systematic approach and the combined use of curvilinear abdominal sector transducers (2–6 MHz) and high-resolution linear transducers (9–12 MHz) for gastrointestinal US may overcome some of these limitations [13]. High-resolution transducers allow visualization of the intestinal wall layers, adjacent mesentery, vessels, and mesenteric lymph nodes. The wall thickness, filling status and peristalsis are assessed with US. The thickness of the wall of small intestine is up to 2 mm. Tumors typically present as segmental wall thickening of the intestinal wall, intraluminal polypoid or nodular masses, or extraluminal masses. Depending on the stage of the tumor, the stratification is usually lost, or individual layers are thickened. The transition to the normal intestinal wall with five layers can be seen at the edge of the tumor. In the area of wall thickening, the lumen reflex is usually narrowed. Luminal obstruction results in dilatation and sometimes hypertrophy of the muscle layer in the proximal intestine. If the small intestine is increasingly filled and shows hyperperistalsis, a pathological intestinal wall finding should be looked for further distally (Figure 2). The localization and ultrasound morphology of the intestinal wall thickening depends on the underlying neoplasia. In adults, intussusceptions can be a concomitant phenomenon of small intestine tumors [40] (Figure 3). Intraluminal gas and intestinal contents can lead to artifacts in US (Figure 4).

In a study of Fukumoto et al. [41] in 2009, all 371 patients in 2003–2008 were consistently screened with 3.5–7.0 MHz ultrasound transducers prior to VCE and/or double balloon enteroscopy (DBE). 92 had a small intestine tumor on VCE and/or DBE. The overall lesion detection rate by US was 25.0% (sensitivity 26.4%; specificity 98.6%). While only 1.8% of tumors < 20 mm were detected by US, this was possible for 59.5% of tumors ≥ 20 mm. Five lesions were detected in US without correlation in VCE and DBE. None of the laterally spreading tumors (granular type) were detected in the US, but all circular ulcerating tumors. One GIST in the lesser pelvis could also not be detected, as the lesser pelvis is not or only partially visible on the US. The authors concluded that US is considered a useful method for detecting small bowel lesions of 20 mm or more and for avoiding unnecessary VCE examinations, especially when associated with the risk of capsule retention.

In a retrospective study by Fujita et al. [42] in 2015, 558 patients with 97 small intestine tumors underwent US with 3.5–7.5 MHz transducers before VCE and/or BAE in 2007–2013. Sensitivity and specificity of US in detecting small intestine tumors were 53.1% and 100%, respectively. For tumors larger than 20 mm, the detection rate was higher (91.7%): The detection rate of submucosal tumors larger than 20 mm was 85.7% and that of partial and circumferential ulcerative tumors larger than 20 mm was 96.9%. There were significant restrictions for tumors < 10 mm. US was able to detect only 2.6% of small intestine tumors < 10 mm. The small intestine tumors detected by US (mean 33.2 mm) were significantly larger than those undetected by US (mean 8.7 mm). Of the small intestine tumors that remained undetected by US, 91.3% were benign tumors [42]. Table 1 summarizes the factors that influence the detection of small intestine tumors.

However, ultrasound technology has evolved with better B-mode resolution and high-resolution linear transducers. Color Doppler imaging (CDI) and contrast enhanced US (CEUS) are helpful in the description and differentiation of lesions.

Examination of the small intestine requires experience, a systematic approach, the consistent use of high-resolution, high-frequency linear transducers, and sufficient time. Table 2 compares the strengths and weaknesses of ultrasound with those of CT and CT/MR enterography.

### 4.6. CEUS

Contrast-enhanced ultrasound (CEUS) is performed as a low mechanical index (MI) procedure using special ultrasound contrast agents (UCA) (SonoVue^®^, Bracco Suisse SA, Geneva, Switzerland; Sonazoid^®^ GE Healthcare AS, Oslo, Norway, Daiichi-Sankyo), [44,45]. When CEUS is used extrahepatically, an arterial phase (0–30 s) and a venous phase (30–120 s) are differentiated. The first contrast signals are usually visible in the gastrointestinal tract 10–20 s after injection, predominantly in the submucosa. The maximum enhancement is achieved after 30–40 s [44]. Linear higher-frequency transducers require higher UCA dosages than conventional abdominal sector transducers. Typical applications of CEUS in the gastrointestinal tract include the diagnosis of abscesses versus phlegmons, the assessment of inflammatory activity and detection of intestinal strictures, fistulas and abscesses in Crohn’s disease [46,47,48]. The detection and differential diagnosis of intestinal tumors is not regarded as a standard indication for CEUS. However, in EUS, assessment of vascularization is important for the differential diagnosis of subepithelial tumors and treatment decisions. Study data from contrast-enhanced low MI-EUS showed that GIST and neuroendocrine tumors are hyperenhanced, while lipomas and the majority of leiomyomas are hypoenhanced [49,50,51,52,53,54,55]. Indications for CEUS include the differentiation of non-enhanced lesions (mesenteric cysts, lymphangiomas) from solid tumors. In cases of doubt, CEUS can be used to distinguish between solid tumors and other intestinal contents in the intestinal lumen. CEUS can be used to differentiate between well-hyperenhanced mesenchymal tumors such as GIST and hypoenhanced tumors such as leiomyomas. In cases of very pronounced hypoechogenicity, it can be demonstrated that the tissue is vital. In percutaneous US-guided sampling, CEUS is helpful in distinguishing between vital and necrotic tissue and in obtaining evaluable histological material [56,57,58]. In addition, the intestinal wall and Interenteric vessels can be better differentiated if necessary.

### 4.7. US-Guided Biopsy

Indications and procedures for transabdominal US-guided sampling of lesions of the gastrointestinal tract or abdominal lymphomas are outlined in the EFSUMB INVUS II guidelines [59]. Indications for US-guided sampling of lesions of the gastrointestinal tract are lesions in the small intestine that are not easily accessible by endoscopy, e.g., GIST or lymphomas. Contraindications are operable lesions of the gastrointestinal tract with a high suspicion of malignancy or lesions where surgery is highly likely to be the first-line treatment [59]. The prerequisite is that the lesion can be visualized with light transducer compression. Passage through sections of the stomach and small intestine with 18-gauge needles is generally safe [60]. Nevertheless, intestinal transit should be avoided. Passage through the colon is contraindicated [59]. With CDI, abdominal vessels can be visualized and avoided in the puncture access [58]. In retrospective studies, sensitivity and accuracy for gastrointestinal tract biopsies using core needles ranged between 81 and 99% [59]. Fine needles performed less well, with sensitivity values of 41 to 50% [61]. We prefer the 18-gauge needle. If lymphoma is suspected, the 16-gauge needle is used as an alternative, considering the risk-benefit ratio.

A percutaneous ultrasound-guided biopsy is always justified if immediate surgery is not indicated, a lesion type is expected that does require neoadjuvant (e.g., tyrosine kinase inhibitor pre-treatment with Imatinib in GISTs) or non-surgical treatment (e.g., chemo- and immunotherapy in malignant lymphoma) and if endoscopic biopsy is not feasible. In a study involving 19 children, 20 percutaneous ultrasound-guided biopsies were performed on small intestine masses. Sixteen were lymphomas, mostly large bulky masses resulting from Burkitt lymphoma (55%). Diagnostic adequacy was 100% [62]. In a study involving percutaneous US-guided biopsies of various areas of the GIT in 45 adults, sensitivity, specificity, PPV, NPV, and diagnostic accuracy, including an inadequate biopsy specimen, were 95.1%, 100%, 100%, 66.7%, and 95.6%, respectively [63].

Normal coagulation is a prerequisite for US-guided sampling. In a study of 36 percutaneous US guided sampling of small bowel lesions, there were no complications [64]. However, potentially complications include bleeding and infections due to perforations [65].

## 5. Small Intestine Adenomas

Small intestine adenomas can occur in the entire small intestine, but mostly in the duodenum and the ampulla Vateri. They are divided into non-ampullary adenomas and ampullary adenomas, the latter being not subject of this review. Non-ampullary adenomas are small intestine polyps of dysplastic glandular epithelium with intestinal or pyloric glandular differentiation.

They occur in any part of the small intestine, but mostly in the proximal duodenum. Around 60% of adenomas occur in patients with Familial Adenomatous Polyposis [66]. Sporadic duodenal polyps are rare and mostly incidental findings on gastroscopy, but account for only 5% of patients at most [67]. They are small and often solitary and can be removed endoscopically. In the remaining small intestine, the diagnosis is often only made when larger adenomas cause lumen obstruction. Adenomas are usually flat or small and are easily overlooked during US. In a study of 50,114 patients, 1.02% had duodenal polyps and 0.3% were duodenal adenomas [68]. The detection of adenomas is a domain of endoscopy.

## 6. Intestinal Carcinoma

Adenocarcinomas account for 30–40% of small intestine tumors [1]. They are most often localized in the duodenum and duodenojejunal junction (>50%), followed by the jejunum and ileum. In the distal small intestine, NEN, lymphomas, or metastatic disease should also be considered

Nevertheless, they are rare and account for <3% of cancers of the entire gastrointestinal tract [1]. Ampullary adenocarcinomas are not discussed in this review.

In Europe, 3600 new cases of adenocarcinoma of the small intestine are diagnosed every year and 5300 in the USA [1]. In contrast to colorectal carcinomas, small intestine adenocarcinomas have a higher proportion of poorly differentiated carcinomas [1].

Chronic inflammatory diseases (Crohn’s disease and celiac disease) are associated with an increased risk of adenocarcinomas in the small intestine. The same applies to some genetic syndromes such as familial adenomatous polyposis, Lynch syndrome and Peutz-Jeghers syndrome [1]. In addition, the possibility of multiple small intestine tumors [69] occurring, sometimes in association with multiple primary malignancies (MPMs) [37], should be considered.

*Imaging:* Tumors in the jejunum and ileum are usually circular ring-shaped and lumen-stenosing. In the duodenum, about one third have a polypoid component, but a proportion of cases show plaque-like growth [70]. On US, the adenocarcinoma presents as hypoechoic wall thickening with lumen narrowing. The infiltrative wall process may show further tongue-shaped hypoechoic infiltrations into the surrounding tissue. In the case of lymph node involvement, the mesenteric lymph nodes at the level of the tumor may be enlarged and/or morphologically conspicuous [71,72] (Figure 5 and Figure 6).

Wu et al. [72] described a single adenocarcinoma in the jejunum using CEUS. The tumor is hypoechoic in B-mode, causing wall thickening and irregular borders. In CEUS, the tumor is hypoenhanced. Wu et al. also performed oral contrast enhancement with UCA [72]. The adenocarcinoma in Crohn’s disease is usually located in the distal jejunum or ileum. It is difficult to distinguish a pathological cockade due to a tumor from a transmural inflammatory wall thickening of Crohn’s disease. For diagnosis, clinical, imaging and laboratory data must be combined with a high degree of experience.

## 7. Neuroendocrine Neoplasms (NEN)

NEN of the small intestine belong to the gastroenteropancreatic NEN (GEP-NEN) [73]. The incidence rate for small intestine NEN was 0.78–1.7/100,000 [74,75,76]. The distribution of GEP-NEN varies regionally: in North America small intestine, colon and rectal NEN are most frequent, in Asia, rectal and pancreatic NEN are most common while in Europe, small bowel and pancreatic NEN are most prevalent [73]. The proportion of small intestine NEN among the GEP-NEN in the various studies was 5.6–29% [77,78,79,80,81,82,83,84].

Small intestine NEN can occur in the small intestine and the ampulla Vateri.

Functionally active NET include gastrinoma, somatostatinoma and paraganglioma. Somatostatin-expressing NET and gangliocytic paragangliomas occur almost exclusively in the ampullary region and in the appendix. The vast majority of jejunoileal NET are located in the distal ileum, only about 11% in the jejunum. Occasionally, enterochromaffin cell (EC cell) tumors develop in Meckel’s diverticula. NEC appear in the small intestine almost exclusively in the ampullary region [1,85].

Duodenal and periampullary NEN are usually small (<2 cm) subepithelial lesions in the submucosa. Jejunoileal NEN may be multifocal in about one third of patients. Upper jejunal NEN can be large and locally invasive. Lower jejunal and ileal NEN are mucosal/submucosal nodules with intact or slightly eroded mucosa. Deep infiltration of the muscular wall and subserosa is common in jejunal and ileal NEN [1]. Most duodenal NEN are diagnosed incidentally during gastroscopy as impressions due to submucosal lesions. First and foremost, they must be differentiated from GIST.

Non-functional NETs can be asymptomatic and be detected incidentally on imaging. However, they can also cause abdominal symptoms due to lumen obstruction. Gastrinoma occurs sporadically or as part of multiple endocrine neoplasia type I. 70% are located in the duodenum, with most of the remaining cases occurring in the pancreas [86]. There are isolated case reports from the jejunum [87]. Somatostatinoma is also very rare and occurs predominantly in the duodenum, the ampullary region, and the pancreas [88], while jejunal localization is reported only in single cases [89].

### Ultrasound Imaging of Small Intestine NEN

On US, small intestine NEN appear as small focal hypoechoic wall thickenings with a pseudo nodular appearance or as hypoechoic nodular lesion. They are usually small tumors. Adjacent pathological lymph nodes may appear in the mesentery. The tumors extend from the submucosa and may extend into the mesentery via the lamina muscularis propria [90,91,92,93,94,95,96]. NEN are usually well vascularized. Color Doppler imaging (CDI) can show irregular macrovessels in the tumor. Case reports showed that small intestine NETs are hyperenhanced on CEUS [91,92,94,95,96,97] (Figure 7 and Figure 8). Further diagnostics include CT and DOTATOC-PET.

## 8. Lymphoma, Hematolymphoid Tumors

Hematolymphoid tumors, and especially lymphomas, occur in the digestive system with varying frequency. They are either primary diseases or part of a systemic manifestation. A primary lymphoma of the digestive system has its origin and its main manifestation in the gastrointestinal tract (Table 3). Regional lymph nodes may be involved, but this is not obligatory. The gastrointestinal tract is the most common site for the occurrence of extranodal lymphomas accounting for 30–40% of all extranodal lymphomas. The small intestine is the second most common location in the gastrointestinal tract with 30% involvement after the stomach (50–60%) [2]. In the small intestine, the ileum and the ileocecal region are most frequently affected. Diffuse large B-cell lymphoma (DLBCL) is the most common type of intestinal lymphoma. Data on the other types of lymphoma vary in studies [2,98]. In the duodenum, follicular lymphoma is the most common subtype [99]. Lymphoid neoplasms (with the exception of histiocytic/dendritic cell neoplasms) are classified according to the Lugano classification [2,100].

### 8.1. Diffuse Large B-Cell Lymphoma (DLBCL)

Diffuse large B-cell lymphoma (DLBCL) is characterized by diffuse neoplastic proliferation of large B-lymphocytes. At 55–69%, DLBCL is the most common lymphoma in the gastrointestinal tract. It is the most common extranodal localization of DLBCL with approximately 40% [2,98]. The average age of onset is in the 6th decade of life. Patients typically present with pain, discomfort, or symptoms associated with luminal obstruction, perforation or gastrointestinal bleeding. B symptoms may be present. In cases of multilocular involvement, it may not be clear where the primary localization is. The lymph nodes may be affected [98].

### 8.2. Extranodal Marginal Zone Lymphoma of Mucosa-Associated Lymphoid Tissue (MALT Lymphoma)

Extranodal marginal zone lymphoma is an extranodal low-grade B-cell lymphoma that develops in mucosa or glandular tissue. It has the cytoarchitectural features of mucosa-associated lymphoid tissue (MALT). MALT lymphoma accounts for 7–8% of all B-cell lymphomas, 30–60% of primary gastric lymphomas and 2–28% of primary intestinal lymphomas [2,101,102]. The median age is in the sixth decade of life. Patients’ complaints are often non-specific or depend on the localization. Some patients are asymptomatic.

### 8.3. Follicular Lymphoma

Follicular lymphoma (FL) is a malignant lymphoid neoplasm of follicular-center B cells. Primary FLs of the gastrointestinal tract are rare and account for <4% of primary gastrointestinal lymphomas. The gastrointestinal tract is the most common extranodal manifestation of FLs. Primary digestive tract FLs often occur in the small intestine and colon [2,99]. Multiple nodular/polypoid lesions have been described endoscopically [103]. The mean age of onset is 50 years. FLs may be diagnosed incidentally, but non-specific abdominal symptoms, clinical symptoms of intestinal obstruction or gastrointestinal bleeding may also occur.

### 8.4. Duodenal-Type Follicular Lymphoma

Typical localization is the second portion of the duodenum. As a rule, these are incidental findings during gastroscopy. There are often further manifestations in the Jejunum and ileum [2,99]. This type of lymphoma is characterized by an indolent clinical course, and excellent outcome.

### 8.5. Mantle Cell Lymphoma

Mantle cell lymphoma (MCL) is a mature B-cell neoplasm. Subclinical tumor cell infiltration of the gastrointestinal tract is common. However, most patients present with stage III/IV disease, primarily with lymphadenopathy and bone marrow involvement [2]. Extranodal involvement usually occurs in conjunction with lymphadenopathy. In the gastrointestinal tract, superficial ulcers, large tumor masses and diffuse thickening of the gastrointestinal mucosa can be observed [104,105,106,107]. The median patient age is around 60 years.

### 8.6. Burkitt Lymphoma

Burkitt lymphoma (BL) is a highly aggressive B-cell lymphoma that often manifests itself in extranodal locations or as acute leukemia. The most common localization in the gastrointestinal tract is the ileocecal region. Typical symptoms are vomiting, abdominal pain, obstruction, jejunal intussusception, bleeding and anemia [2,62,108,109,110].

### 8.7. T-Cell Lymphoma

#### 8.7.1. Enteropathy-Associated T-Cell Lymphoma

Enteropathy-associated T-cell lymphoma (EATL) is an intestinal T-cell lymphoma originating from intraepithelial lymphocytes in patients with celiac disease. EATL accounts for <5% of all gastrointestinal lymphomas. All patients have the genetic background of celiac disease [2,101,111]. As a rule, patients are over 50 years old. Symptoms include abdominal pain, diarrhea and weight loss. Manifestations with acute abdomen due to perforation or ileus have also been described.

#### 8.7.2. Intestinal T-Cell Lymphoma NOS

Intestinal T-cell lymphoma (ITCL) NOS is an aggressive malignancy affecting mostly the colon and small intestine [2]. ITCL can be limited to one gastrointestinal organ or have multiple localizations. Dissemination to regional lymph nodes and extra gastrointestinal sites is also possible.

### 8.8. Imaging of Lymphomas

Due to the pronounced hypoechoic wall thickening with compressed central hyperechoic lumen reflex, there are comparative descriptions of the lymphoma manifestations in US as “doughnut sign” [112], “bull’s eye”, ‘target’ or “pseudo kidney” sign [110,113].

A common feature of small intestine lymphoma is a very pronounced wall thickening with marked hypoechogenicity, accompanied by enlarged lymph nodes. However, there are also focal circumscribed polypoid nodular changes or thickening of the mucosa and submucosa, which are mainly detectable endoscopically and are more difficult to delineate with cross-sectional imaging. The appearance of lymphomas in the small intestine depends on the extent of the different subtypes in the intestinal wall. The extent also depends on the stage of the disease (Figure 9 and Figure 10). Görg et al. [114] described the various manifestations of lymphoma in the gastrointestinal tract (including the small intestine) as mucosa-associated, transmural segmental, transmural circumferential, transmural nodular, and transmural bulky disease [114].

In a study of Hasaballah et al. [113] including 21 patients with small intestine lymphoma (90% DBLCL), marked wall thickening (15.6 ± 5.9 mm) with loss of stratification (76%) and high hypoechogenicity were the typical features of the lymphoma manifestations. In 85.7% the lymphoma was unilocular. Further features were dilatation of the intestinal lumen and periintestinal enlarged lymph nodes. US-guided biopsy was performed in 33% of patients and was successful in 100% [113].

Zhang et al. [115] distinguished the mass type, wall thickening type and non-specific signs in 19 intestine lymphomas. The mass type was characterized by solid and cystic-solid hypoechoic lesions with well-defined margins. Among them were pronounced hypoechoic lesions with dorsal enhanced cancellation. On CDI, abundant blood signals were visible in large masses.

The wall thickening type was characterized by thickening of the bowel walls with hypoechoic (including marked hypoechoic) foci and posterior acoustic enhancement. Nonspecific signs included dilatation of the bowel and enlargement of the mesenteric lymph nodes [115].

Cui et al. [116] described four sonographic patterns on B-mode US in 18 patients with small intestine lymphoma. The majority (61%) were DLBCL. The mass type (66.7%) presented as circumferential, marked intestinal wall thickening. The wall was highly hypoechoic with loss of stratification. The central luminal reflex was hyperechoic. The infiltrative type (5.6%) was a segmental thickening of the bowel wall without visible mass formation with a bowel wall thickness of 2.6 cm in a single case. The mesenteric type (22.2%) was characterized by multiple hyperechoic heterogeneous mesenteric masses with or without involvement of the adjacent bowel wall. The lesion sometimes encapsulated mesenteric vessels without stenosis of the vessel lumen. The mixed type (5.6%) showed a combination of two patterns [116].

On CEUS all small intestine lymphomas showed arterial enhancement (hyperenhancement in 17 of 18 cases) followed by venous washout. Tumor necrosis was observed in 61% of cases, which occurred more frequently in aggressive subtypes than in indolent subtypes. There was no correlation between tumor size and necrosis [116].

Standard imaging is performed using CT, MRI and PET-CT. US-guided sampling can be used for histological confirmation. Follow-up checks for treatment response can include US.

## 9. Mesenchymal Tumors

The most common mesenchymal tumor in the gastrointestinal tract is the gastrointestinal stromal tumor (GIST) with an overall incidence estimated at 15/100,000 people [117,118]. GIST amount to 1–3% of all gastrointestinal neoplasms [119,120]. The small intestine (30%) is the second most common location after the stomach (50–70%) [3,121]. All other mesenchymal tumors except GIST are very rare in the small intestine. According to the WHO classification of soft tissue and bone tumors, there are four categories of tumor behavior: benign, intermediate (locally aggressive), intermediate (rarely metastatic), and malignant [3,122] (Table 4).

### 9.1. Gastrointestinal Stromal Tumor

GIST are the most common mesenchymal tumors in the gastrointestinal tract and in the small intestine. They are also referred to as SIST (small intestinal stromal tumor) [123]. In a study by Miettinen et al. of 906 small bowel GIST, the majority manifested with gastrointestinal bleeding followed by acute abdomen, only 18% were incidentalomas [121]. Gastrointestinal bleeding was also the most common clinical sign of GIST in other studies [123,124]. In rare cases, a GIST can also manifest itself with an upstream intussusception of the small intestine [125]. In a study of 75 small intestine GIST, the most common site was the jejunum (57%), followed by the ileum (32%) and duodenum (11%) [123].

GIST can be associated with several genetic syndromes. Among these, patients with neurofibromatosis type 1 may have multiple GIST in the small intestine [3,121,126].

In the study of 75 small intestine GISTs with a size of 0.9–20 cm, 51% of them were <5 cm, only 30% were detected by US. In contrast, dual-source CT diagnosed 87.3% of the tumors [123]. In another study of 32 patients with small intestine GIST and a mean tumor size of 5.3 cm, only 28% underwent an US examination. In eight of nine cases, the tumor could be identified but not assigned to an organ. This was very similar on MRI. An assignment to the intestinal wall was achieved by CT in 54% and by CT angiography in 71% [124].

GIST originate from the myenteric plexus in the muscularis propria and, therefore, are usually associated to the 4th wall layer (lamina muscularis propria). In an early US study [127] of cases with unexplained gastrointestinal bleeding, careful examination with 7.5 MHz transducers revealed mesenchymal tumors of a size of 3–6 cm. The positional variability caused by the motility of the small intestine was striking. Intermittent intussusceptions were also conspicuous [127].

Precise descriptions of subepithelial intramural masses including GIST are known from endoscopic ultrasound (EUS) [128]. Despite the lower resolution and proximity, the findings known from EUS can generally be transferred to US. However, in the small intestine, it is much more difficult to classify the layers, as the healthy wall is only up to 2 mm thick. GIST can develop into the lumen of the gastrointestinal tract or be located extraluminally. On EUS as well as on US, small GIST are generally round, homogeneous, and hypoechoic compared to submucosal layer, but isoechoic to the Lamina muscularis propria. On CDI, they show macrovessels. Larger GIST may be lobulated and heterogeneous, show necroses as hypoechoic or nonechoic liquid areas as well as calcifications. Air inclusions in the tumor are an indication of deep ulceration of the tumor surface with central necrosis and connection to the gastrointestinal tract lumen [55] (Figure 11 and Figure 12).

In a multicenter study, 98% of GIST of all locations, including the small intestine (no specific location), showed arterial hyperenhancement on contrast enhanced low mechanical index EUS (CELMI-EUS) with SonoVue. Some demonstrated non-enhanced areas as evidence of necrosis [55].

### 9.2. Inflammatory Myofibroblastic Tumor (IMFT)

On US, small intestine IMFT is described as well-defined and hypoechoic. The tumor may appear heterogeneous with hyperechoic parts and dorsal attenuation as well as small calcifications [129]. On CEUS with SonoVue, the IMFT demonstrated contrast enhancement in the arterial phase but slightly lower intensity than the adjacent small bowel wall, followed by central washout with marginal enhancement [129].

### 9.3. Inflammatory Fibroid Tumor (Vanek’s Tumor)

The only US description of an ileal IFP is as a clearly defined “target-like” lesion, which led to ileo-ileal intussusception [130] (Figure 13).

### 9.4. Desmoid Tumors

Desmoid tumors are locally aggressive mesenchymal tumors with a tendency to local recurrence. They are generally very rare. The incidence of desmoid tumors is 0.03% of all neoplasms and less than 3% of all soft tissue tumors [131]. There are associations with familial adenomatous polyposis, Gardner syndrome and mutations in the adenomatous polyposis coli (APC) gene [132]. Small intestine manifestations are very rare without any description of US or CEUS features in the few case reports [132] (Figure 14). Apart from musculoskeletal US descriptions, there are descriptions of US characteristics including CEUS and EUS of desmoid tumors in solid abdominal organs [133]. Desmoid tumors appear as well-circumscribed, homogeneous masses on CT [132].

### 9.5. Lymphangioma

Lymphangiomas are very rare benign tumors of the gastrointestinal tract and can occur in the intestine, the gall bladder and the pancreas [3]. However, they are more likely to be vascular malformations that are filled with lymph. Lymphangiomas consist of different sized spaces that contain lymph. They exist in three variants: capillary, cavernous and cystic variants [134]. Children are usually affected. However, lymphangiomas can also be diagnosed in adulthood if they cause intussusception or other acute obstructive symptoms due to the increase in size [135,136,137]. However, as with other intestinal tumors, lymphangioma can manifest with gastrointestinal bleeding. US can be used to detect and localize the most anechoic lesions and determine their cystic appearance [134,135] (Figure 15). On CT, the tumors appear as homogeneous, non-enhancing lesions with variable attenuation values depending on whether the fluid is chylous or serous [134].

### 9.6. Lipoma

Lipomas are benign, smoothly bordered oval/round lipomatous lesions that usually originate in the submucosa. They appear homogeneous and hyperechoic on EUS and US (Figure 16). They are rare in the small intestine.

### 9.7. Schwannoma

Schwannomas are tumors that develop from Schwann cells in the sheaths of peripheral nerves. They are very rare in the small intestine, and only a few reports on schwannomas of the small intestine have been published [138,139,140]. Schwannomas exhibit two microscopic growth patterns, Antoni A (hypercellular component) and Antoni B (hypocellular component). In contrast-enhanced CT scans, Antoni A areas show heterogeneous enhancement. Antoni B areas are hypocellular and show a homogeneous pseudocystic appearance on CT without significant contrast enhancement. In addition, true cystic degeneration may occur in Antoni B areas due to vascular thrombosis and necrosis. Due to the varying extent of Antoni A and B areas, the appearance of schwannomas in imaging can vary considerably [141]. Schwannomas can be solid, cystic-solid, encapsulated, and clearly defined (Figure 17). We are not aware of any descriptions of small intestine schwannomas in the US or CEUS. In other organs, schwannomas show enhancement of the capsule and septa in CEUS [142].

## 10. Metastases in the Small Intestines

Metastases of other tumors can occur in the small intestine. These include cutaneous melanomas, breast carcinomas, lung carcinomas, renal cell carcinoma, thyroid cancer, ovarian tumors, testicular tumors and head and neck squamous cell carcinoma [4,143,144,145,146,147,148]. In a study of more than 5000 patients, melanomas (66% of all small bowel metastases) were the most common secondary small intestine tumors [32]. In a study of 193 malignancies in the small intestine over a period of 20 years, 5.7% were metastases [146]. The metastases were located in both the jejunum and ileum [146]. Small bowel metastases of head and neck squamous cell carcinoma were mostly located in the ileum (58%), also jejunum (25%), and less frequently in the duodenum (15%) [143]. Clinical manifestations of small intestine metastases are gastrointestinal bleeding, abdominal discomfort up to acute abdomen in connection with intestinal lumen obstruction and also intussusception [146]. They manifest themselves as wall thickenings or intramural nodules in the wall of the small intestine (Figure 18 and Figure 19).

The typical appearance of the various small intestine tumors and indirect indications are summarized in Table 5. Differential diagnoses of tumor-related wall thickening in the small intestine include inflammatory wall changes in Crohn’s disease and duodenal wall hematoma (Figure 20, Figure 21 and Figure 22). If wall thickening does not regress in long-standing Crohn’s disease under anti-inflammatory therapy and there are no explicit clinical and paraclinical signs of inflammation, the development of a tumor on the basis of Crohn’s disease must always be considered.

## 11. Future Directions and Artificial Intelligence (AI)

There is experience with the use of AI in US for diffuse liver disease and focal liver lesions [149,150]. The integration of CEUS in the differentiation of liver lesions has been investigated in studies [150,151]. The main focus of AI in gastroscopy is on cancer detection, determining the depth of cancer invasion, and predicting pathological diagnoses. In the field of VCE, the automated detection of bleeding, ulcers, tumors, and various small bowel diseases is being investigated. In colonoscopy, prospective studies are being conducted on the automated detection and classification of colon polyps and inflammation in IBD [152]. The primary objective of this publication was to raise awareness among examiners of the detection of small intestine tumors using US. This requires excellent training and experience in US, systematic and thorough examination of the small intestine, additional scheduling for the examination, and appropriate US technique. We would like to see AI assist in the detection of small intestinal tumors, similar to endoscopy. This would be the most important factor. Further steps would include the differential diagnosis of small intestinal tumors using multiparametric ultrasound, including B-mode, CDI, and, if necessary, elastography and CEUS. There is very little data available for CEUS of small intestine tumors. Therefore, data that reveal, for example, characteristic enhancement patterns should be evaluated in multicenter studies. In our opinion, there is insufficient experience with microvascular imaging for small intestinal tumors. A further step would be, for example, the differential diagnosis (before histology) by verifying vascularization using microvascular imaging, CEUS, and qualitative and qualitative dynamic CEUS.

## 12. Conclusions

Small intestinal tumors are rare. Diagnostic management of patients with unexplained anemia, gastrointestinal bleeding, abdominal complaints or other symptoms indicating malignant disease that cannot be clarified by gastroscopy and ileocolonoscopy, should undergo examination for small intestine tumors. In Figure 23, Figure 24 and Figure 25, we suggest the potential diagnostic steps for the evaluation of occult bleeding, suspected small bowel obstruction, and suspected new obstruction in IBD/Crohn’s disease.

In literature, initial diagnosis of small intestine tumors is most often made by CT. An endoscopic examination of the jejunum and ileum is extensive and not completely possible and only intended for specific problems. CT enterography and MR enterography are more complex examinations, and CT is associated with radiation exposure. VCE is expensive and associated with false negatives in the context of subepithelial lesions.

While small intestine tumors < 10 mm tend to be overlooked, gastrointestinal US has a good detection rate for larger tumors > 20 mm. In our opinion, it is important to allow sufficient time for an examination of the small intestine and to perform the examination systematically using high-resolution transducers. US can be used to detect wall thickening, surrounding infiltrations, and lymph nodes. Indirect signs such as luminal dilation of the proximal intestinal segment, muscular hypertrophy, or intussusception can be visualized with US. With CEUS, additional information can be obtained about vascularization, and necrotic tumor areas can be delineated. Detection and characterization of small intestine tumors requires special knowledge and training in gastrointestinal ultrasound applications. The detection of small intestine tumors on ultrasound allows the targeted use of further complex diagnostic procedures for further characterization and to obtain a histologic diagnosis, occasionally also through US-guided sampling.

## 13. Take Home Messages

Small intestine tumors are rare but clinically significant, especially in patients with unexplained anemia, GI bleeding, or abdominal pain not clarified by endoscopy.Transabdominal US is an underutilized but valuable tool for the detection of small bowel tumors, particularly those ≥20 mm in size.High-resolution transducers and systematic scanning techniques are essential to visualize bowel wall stratification, segmental thickening, and indirect signs like proximal dilation or intussusception.Multiparametric US, including CDI and CEUS, enhances lesion characterization by assessing vascularity and necrosis, especially in GISTs and neuroendocrine tumors.Ultrasound-guided biopsy is a reliable diagnostic option for accessible tumors not amenable to endoscopic sampling, particularly in lymphomas and large stromal tumors.US can guide further diagnostic decisions, reducing unnecessary CT, MRI, or capsule endoscopy in selected clinical scenarios—provided sufficient time, expertise, and high-quality equipment are available.We advocate for a broader use of multiparametric ultrasound in the diagnostic workup of patients with suspected small intestinal tumors.

## Figures and Tables

**Figure 1 healthcare-13-02776-f001:**
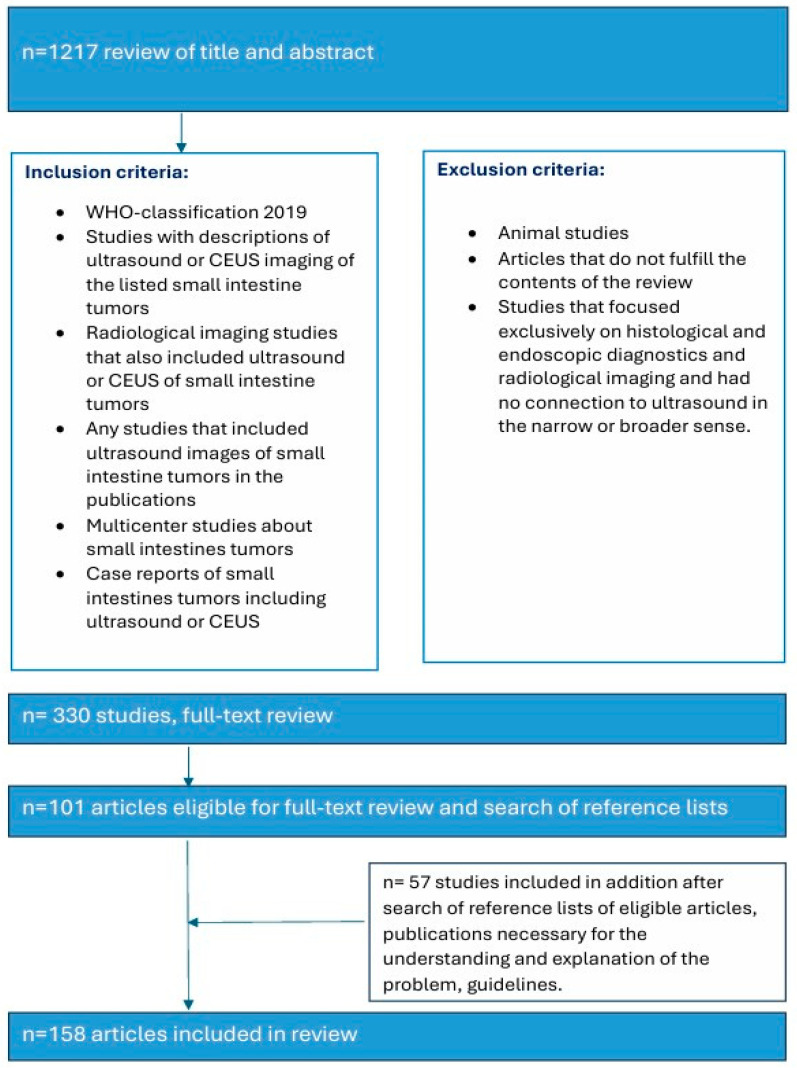
Search strategy.

**Figure 2 healthcare-13-02776-f002:**
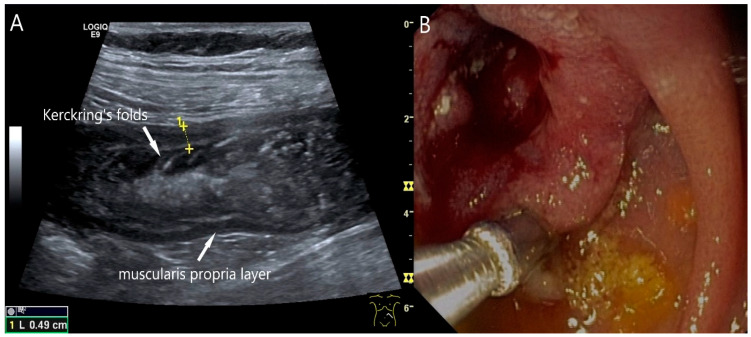
Adenocarcinoma of the Jejunum. Unclear cramp-like abdominal pain, inconclusive gastroscopy and ileocolonoscopy. US shows dilation of the jejunum with active peristalsis. The wall is thickened to 4.9 mm (between the markers). The lamina muscularis propria (white arrow) is prominent, hypertrophic with visibility of the radial and longitudinal muscle layer with echogenic collagen fibers in between (**A**). Although the US findings did not reveal the cause, they prompted further diagnostics. Push enteroscopy detected a circumferential growing and stenosing polypoid tumor (**B**).

**Figure 3 healthcare-13-02776-f003:**
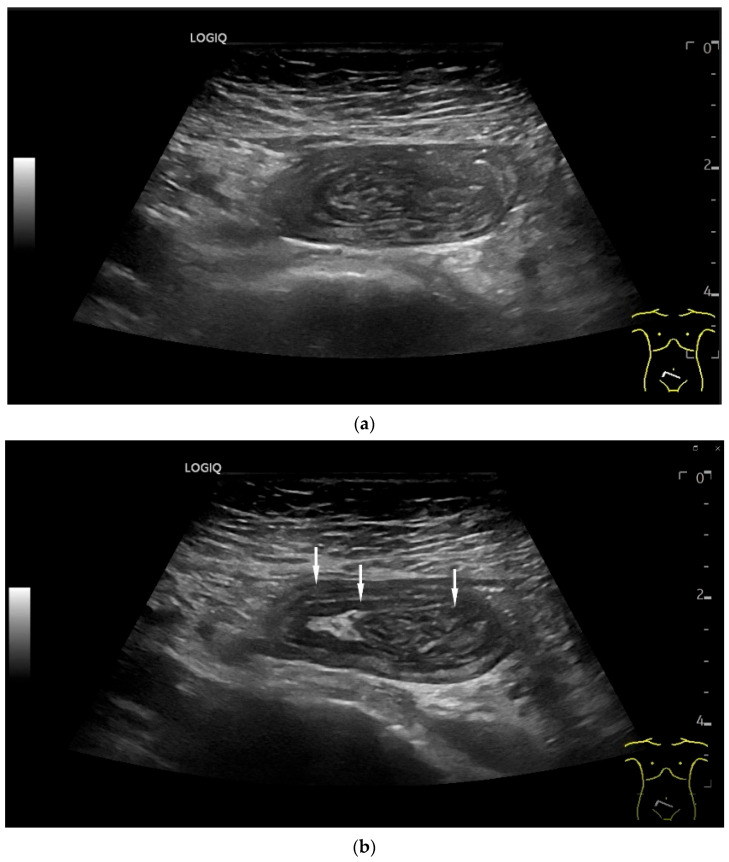
Intussusception. In the ileum, the intestine is inverted. Instead of the usual five layers, there are many onion-skin-like layers (**a**). The intestine is folded in on itself several times. The wall is marked with arrows (**b**).

**Figure 4 healthcare-13-02776-f004:**
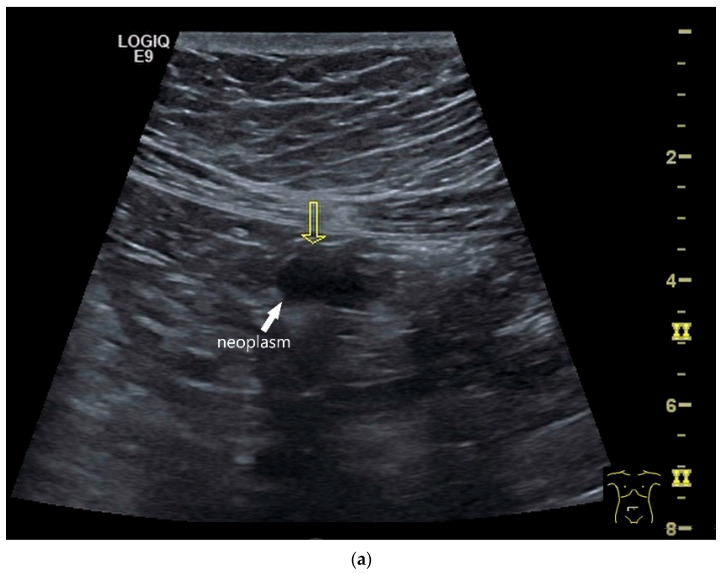
NEN in the ileum: an approximately 14 mm hypoechoic nodular tumor is visible (**a**). Subsequently, peristalsis reveals a hypoechoic wall thickening (W) and hyperechoic luminal reflex (L) (**b**) and, finally, causes the tumor to be obscured by the luminal contents and dorsal artifacts (**c**). The region of the tumor is marked with a yellow arrow in all three images.

**Figure 5 healthcare-13-02776-f005:**
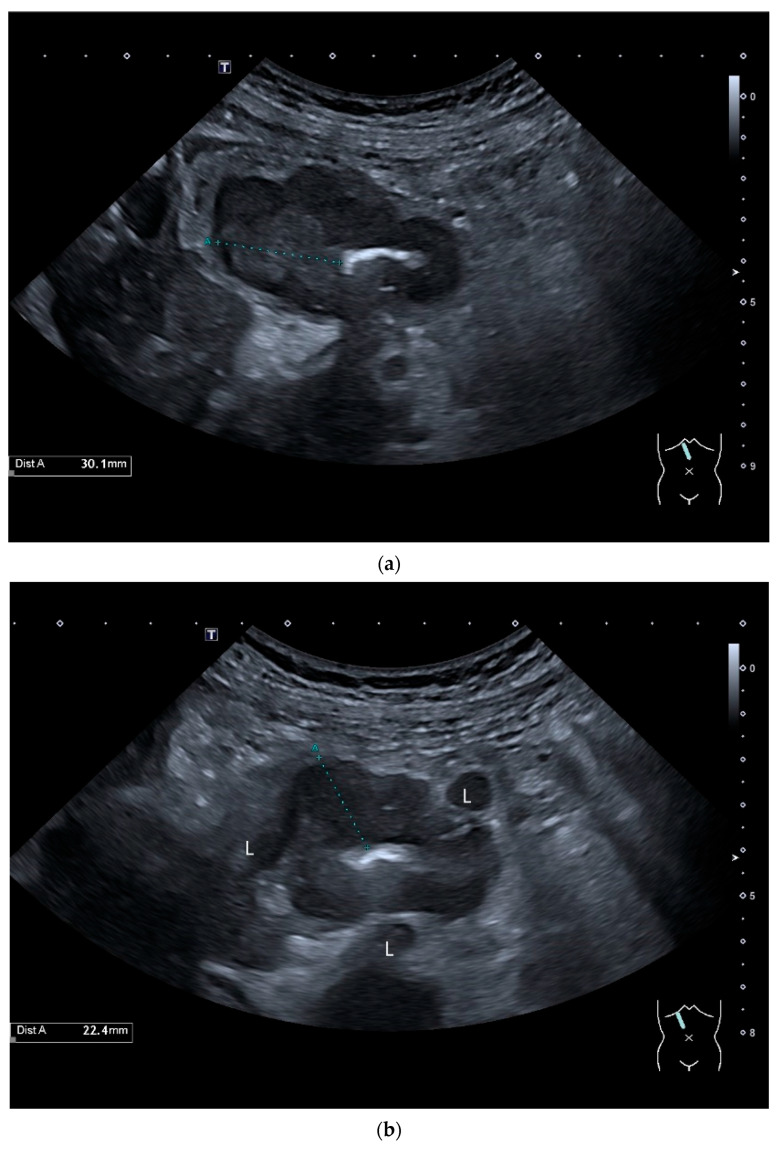
Adenocarcinoma in the duodenum. Significant hypoechoic wall thickening (between the markers) with narrowed lumen reflexes (**a**). The tumor has a slightly polycyclic border on the outside (**a**,**b**). Several tumor-suspicious, round, hypoechoic lymph nodes (L) are visible in the paraduodenal region (**b**).

**Figure 6 healthcare-13-02776-f006:**
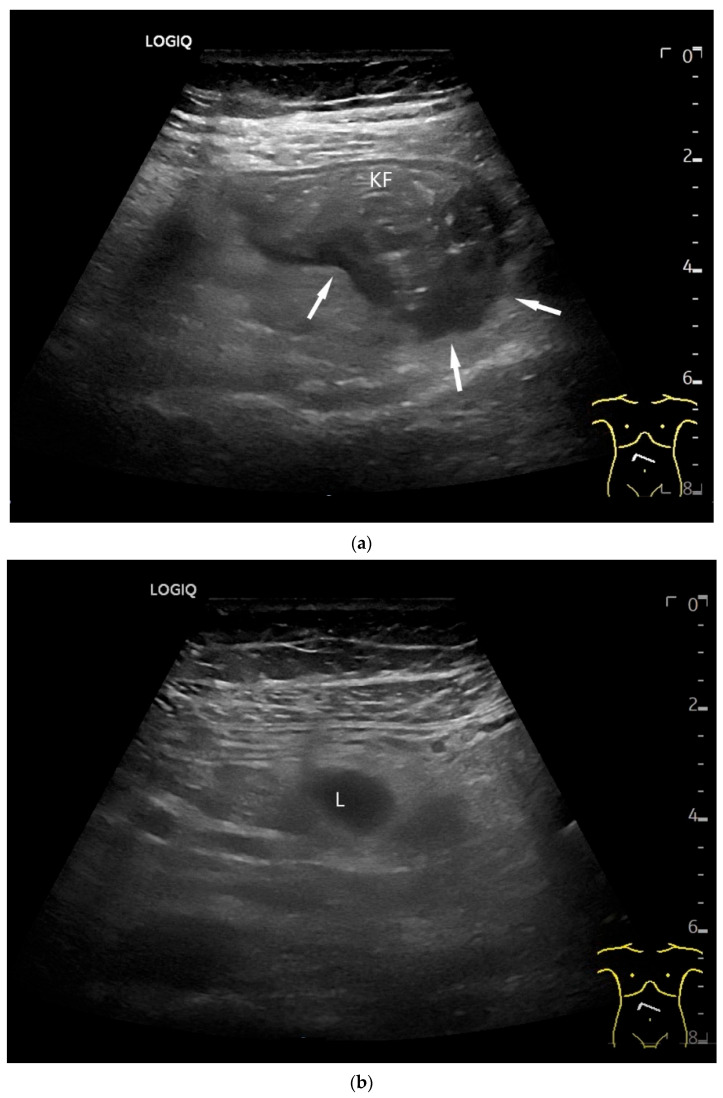
Jejunal Adenocarcinoma. Segmental hypoechoic wall thickening (arrows) in the jejunum discovered during anemia diagnostics. Kerckring folds are faintly visible (KF) (**a**). Adjacent to this is a large round hypoechoic lymph node (L). The surrounding area shows hyperechoic changes (**b**).

**Figure 7 healthcare-13-02776-f007:**
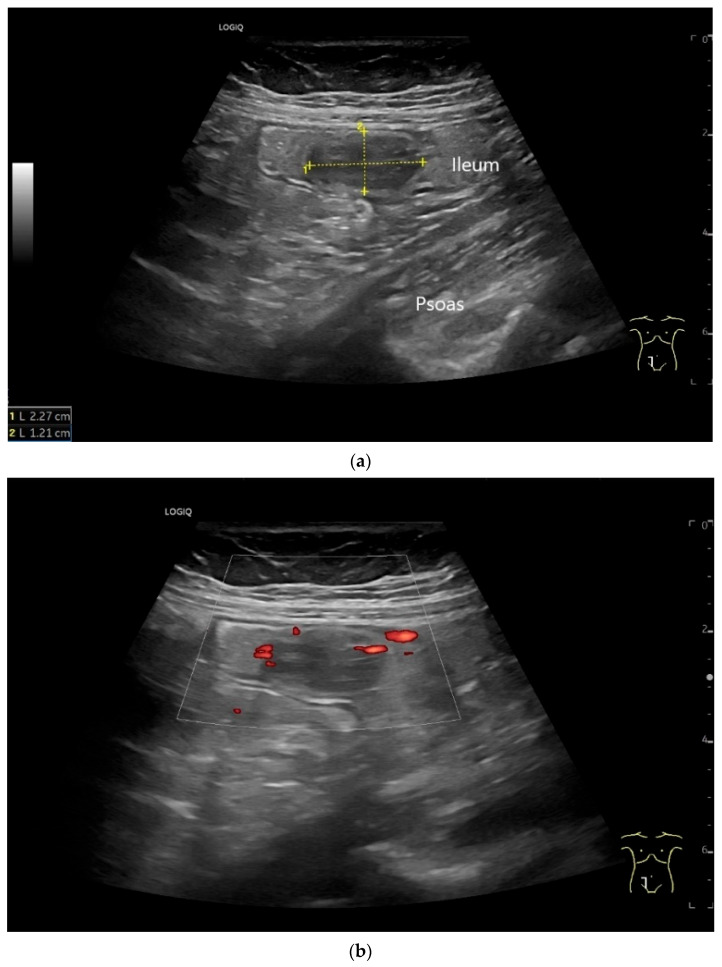
NET in the ileum. An oval polyp-like lesion is visible in the ileum (**a**). Small vessels are visible on power Doppler, indicating that the tissue is solid (**b**). In CEUS with 2.4 mL SonoVue (linear transducer 2–9 MHz), the tumor capsule shows enhancement. The tumor is marked with white arrows. At 22 s, vessels can be distinguished in the tumor, but there is no complete enhancement of the tumor. The ileum wall is also enhanced (yellow arrow) (**c**). Even at the end of the arterial phase at 34 s, the tumor is not completely enhanced. Two macro vessels are marked with an arrow. Distally, they show irregular vessel branches (**d**). The tumor was not located directly behind the ileocecal valve. Based on the US findings, the endoscope was inserted 20 cm into the terminal ileum, where the tumor was then discovered.

**Figure 8 healthcare-13-02776-f008:**
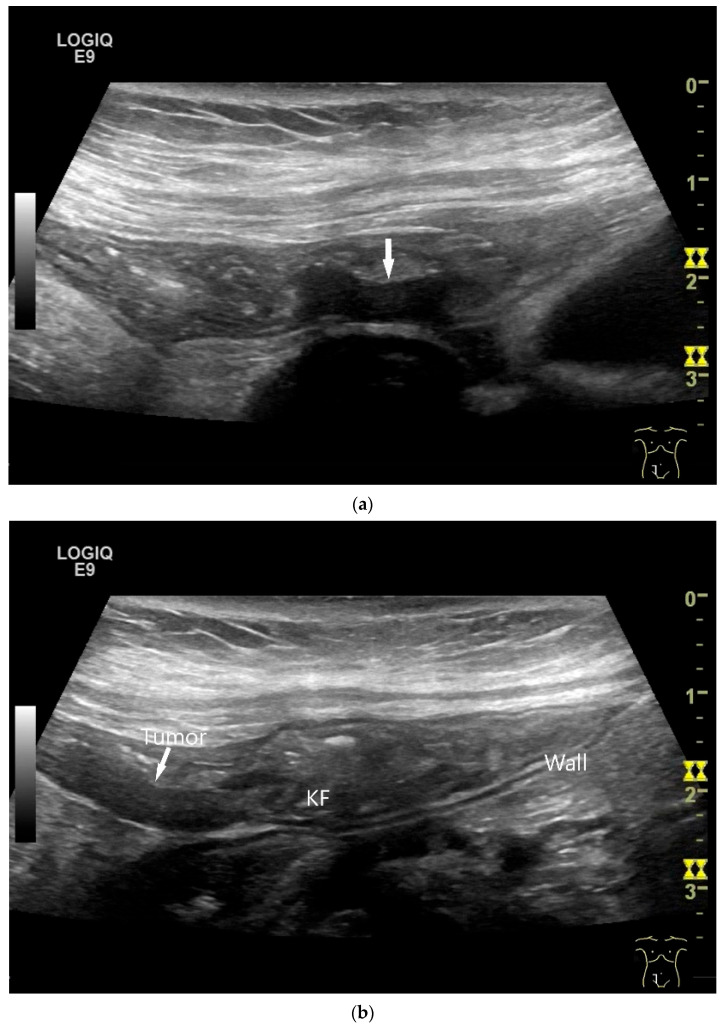
NET in the ileum. In the terminal ileum, there is a slight wall thickening <5 mm (arrow), but with loss of stratification (**a**). The tumor (arrow) changes position with peristalsis while the transducer position remains the same. With the 9 MHz linear transducer, the five layers of the unremarkable wall and Kerckring folds (KF) are visible (**b**). Small vessels can be distinguished in power Doppler (**c**).

**Figure 9 healthcare-13-02776-f009:**
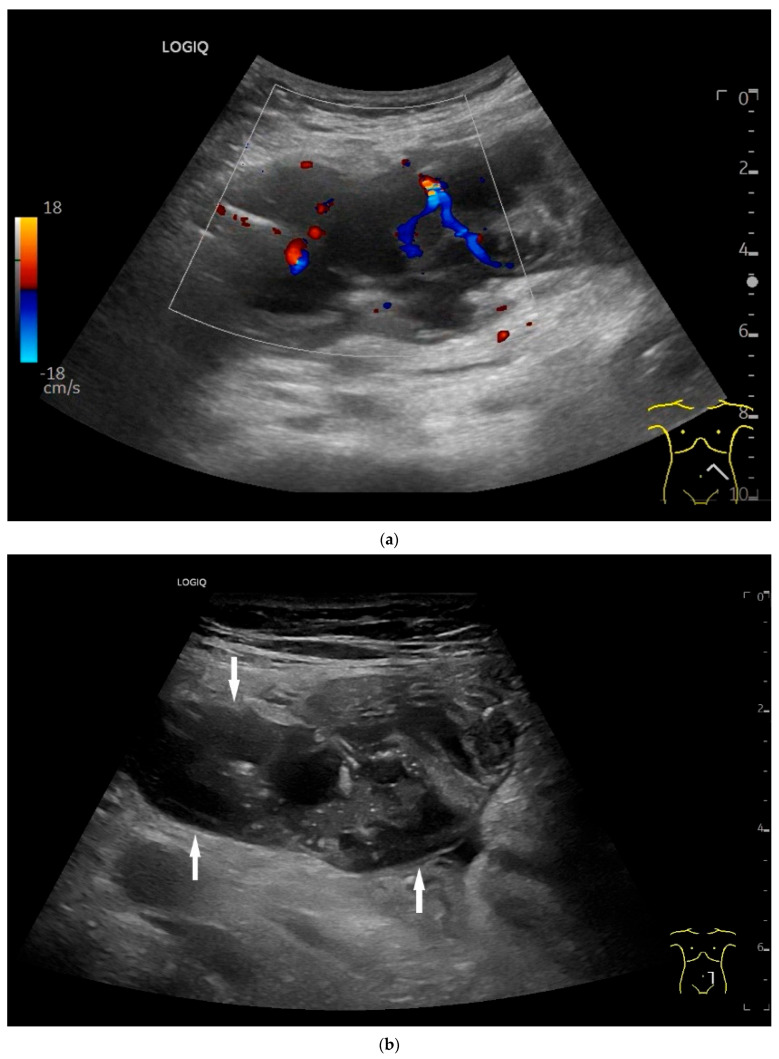
Burkitt-Lymphoma. In a patient with weight loss, increased abdominal circumference, anemia, and physical weakness, the initial sonographic examination reveals an extensive tumor with intense hypoechogenicity and a connection to the jejunum in the left mid-abdomen. Despite its pronounced hypoechogenicity, a feeding vessel on CDI indicates a solid character (**a**). The tumor significantly thickens the wall and is intensely hypoechoic. The arrows point to the multisegmental hypoechoic wall thickenings (**b**). Normal wall structures are still visible (W), and the tumor extends beyond the wall (arrow) (**c**). In addition to delicate Kerckring folds (KF), there are significantly polypoid thickened KF with pronounced hypoechogenicity (**d**). The thickening of KF is very extensive (**e**). Wall thickening was found also in the stomach and colon and allowed endoscopic biopsy to establish the diagnosis.

**Figure 10 healthcare-13-02776-f010:**
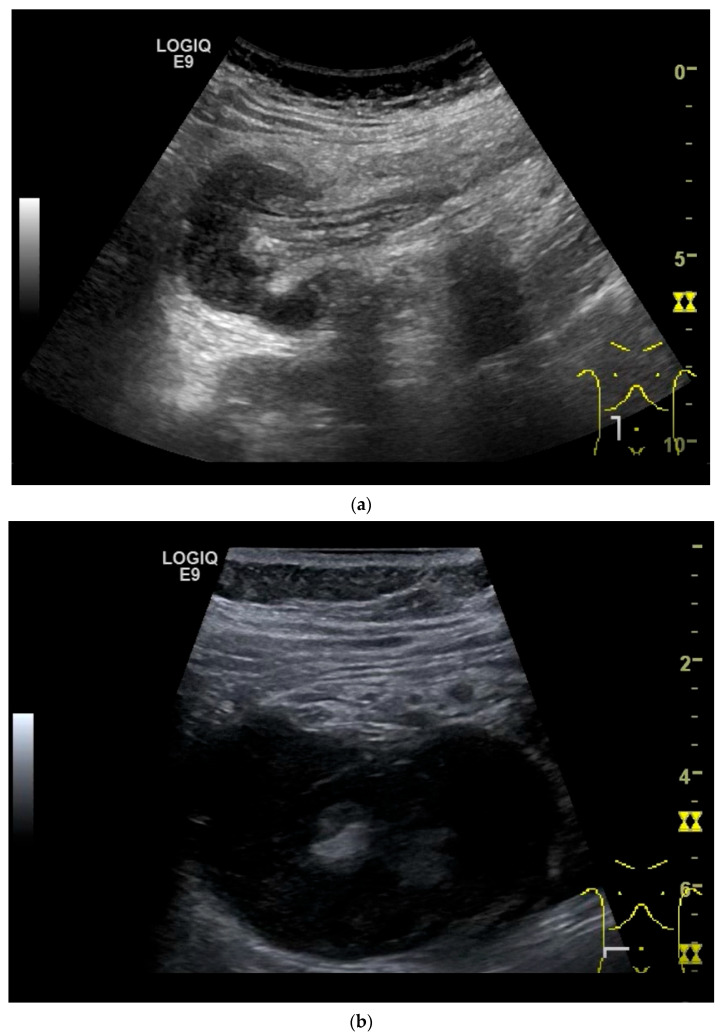
Diffuse large B-cell lymphoma (DLBCL). In the right mid-lower abdomen, a large, smoothly circumscribed mass is present around the ileum (“pseudo kidney sign” (**a**)). Using a high-resolution linear transducer, the mass appears smoothly circumscribed and almost anechoic (**b**). The ileum runs centrally, and the echogenic wall is clearly defined (**c**). This wall does not merge into the mass. Since the mass is almost anechoic, it is also difficult in different transducer positions to distinguish whether it is a liquid lesion or a solid mass (**b**,**c**). CEUS with 2.4 mL SonoVue (9 MHz linear transducer) shows homogeneous arterial enhancement (**d**), but the intensity decreases with time in the venous phase (**e**). The ileal wall enhancement centrally within the mass is more intense than that of the tumor (**d**,**e**).

**Figure 11 healthcare-13-02776-f011:**
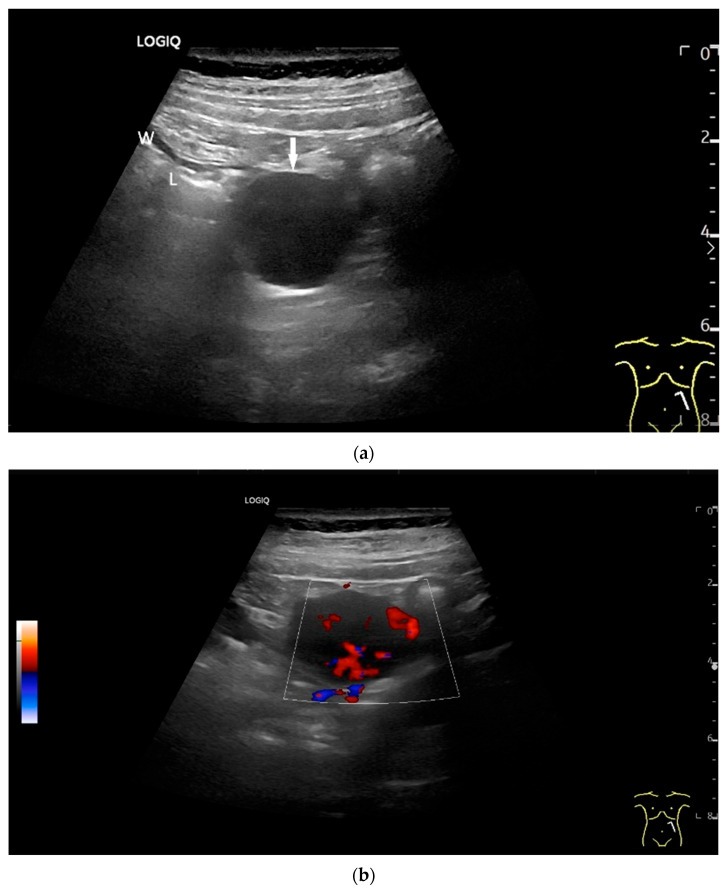
GIST. A 30 mm large, very hypoechoic, almost anechoic round mass is visible in the left upper abdomen. This is located in the jejunum; the wall (W) and lumen (L) are visible (**a**). Macro vessels can be distinguished on Power Doppler, demonstrating that the lesion is solid and not cystic (**b**). On CEUS with 2.4 mL SonoVue (linear transducer 9 MHz), a small wheel-spoke-like vascular branching is visible at the margin (arrow) (**c**) with centrifugal enhancement (arrow) (**d**). Hyperenhancement is heterogeneous in the early arterial phase (**e**,**f**) and becomes homogeneous in the later course of the arterial phase (**g**). The extent of the heterogeneously enhanced tumor is marked with arrows (**e**). The intensity of the enhancement decreases during the first minute. The tumor is marked with arrows (**h**). Jejunal segment resection revealed the histology of an epithelioid GIST.

**Figure 12 healthcare-13-02776-f012:**
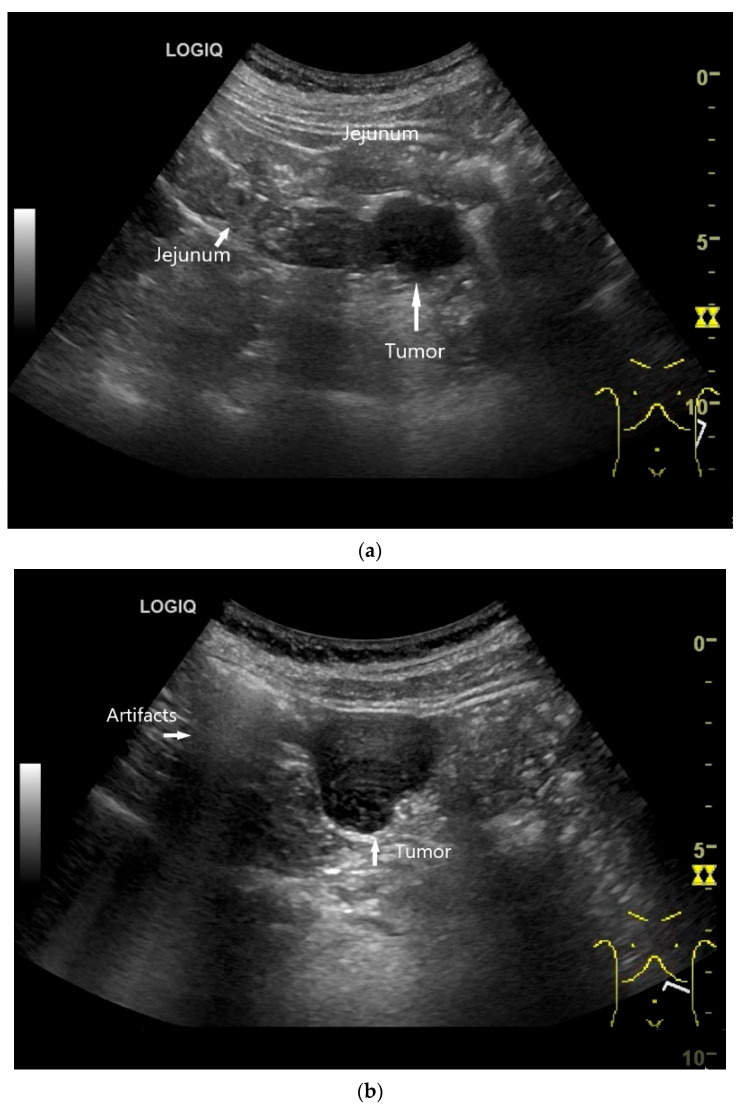
GIST. Incidental finding of a 35 mm, slightly polycyclic, homogeneous hypoechoic mass in the left upper abdomen (**a**,**b**). This changes position with the movements of the small intestine. In CEUS with 1.2 mL SonoVue using the abdominal sector transducer (1–5 MHz), the mass shows homogeneous hyperenhancement in the arterial phase (**c**). The intensity then decreases (**d**). The tumor is marked with arrows in CEUS.

**Figure 13 healthcare-13-02776-f013:**
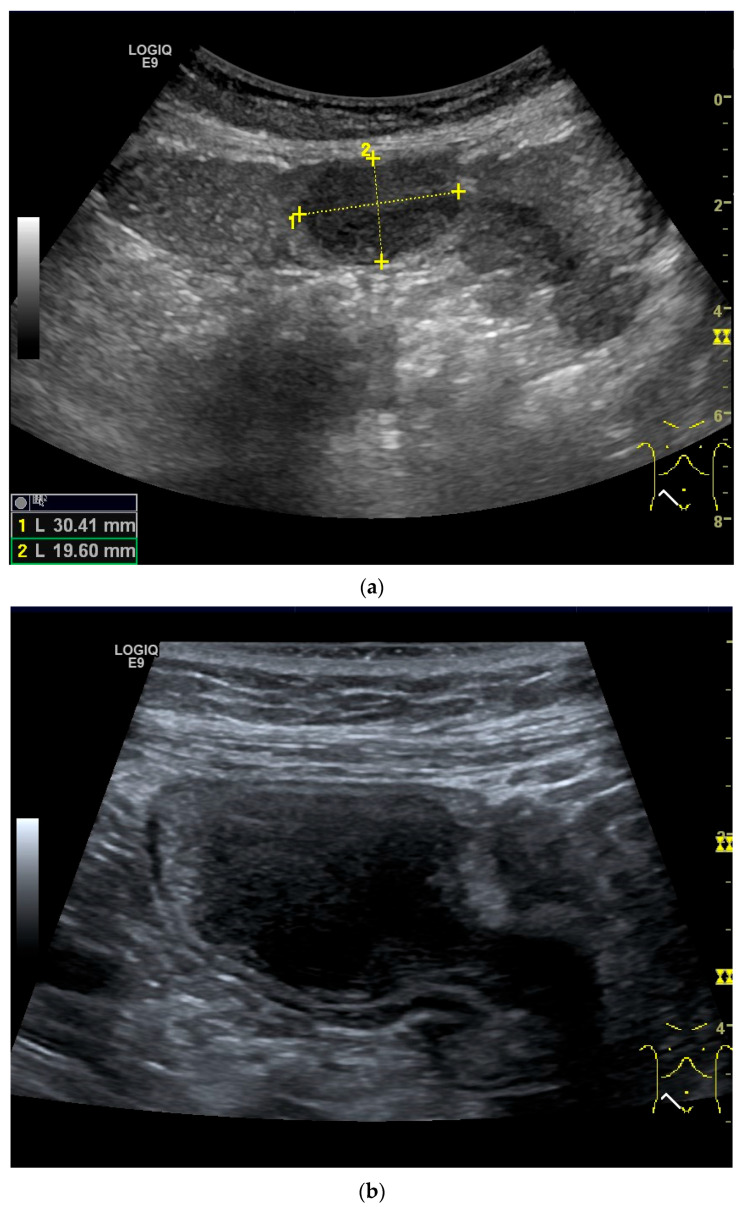
Inflammatory fibroid tumor (Vanek’s tumor). In cases of unclear occult gastrointestinal bleeding, a smooth-edged oval hypoechoic mass measuring up to 30 mm in size can be seen in the terminal ileum. Examination with the abdominal sector transducer 1–5 MHz (**a**) and linear transducer 2–9 MHz (**b**). In power Doppler, a feeding vessel (**c**) and macro vessels in the mass (**d**) are visible. In CEUS with 2.4 mL SonoVue using a 9 MHz linear transducer, hyperenhancement is only visible at the edge in the arterial phase. Few vascular signals are visible in the lesion (**e**). The tumor remains hypoenhanced (**f**).

**Figure 14 healthcare-13-02776-f014:**
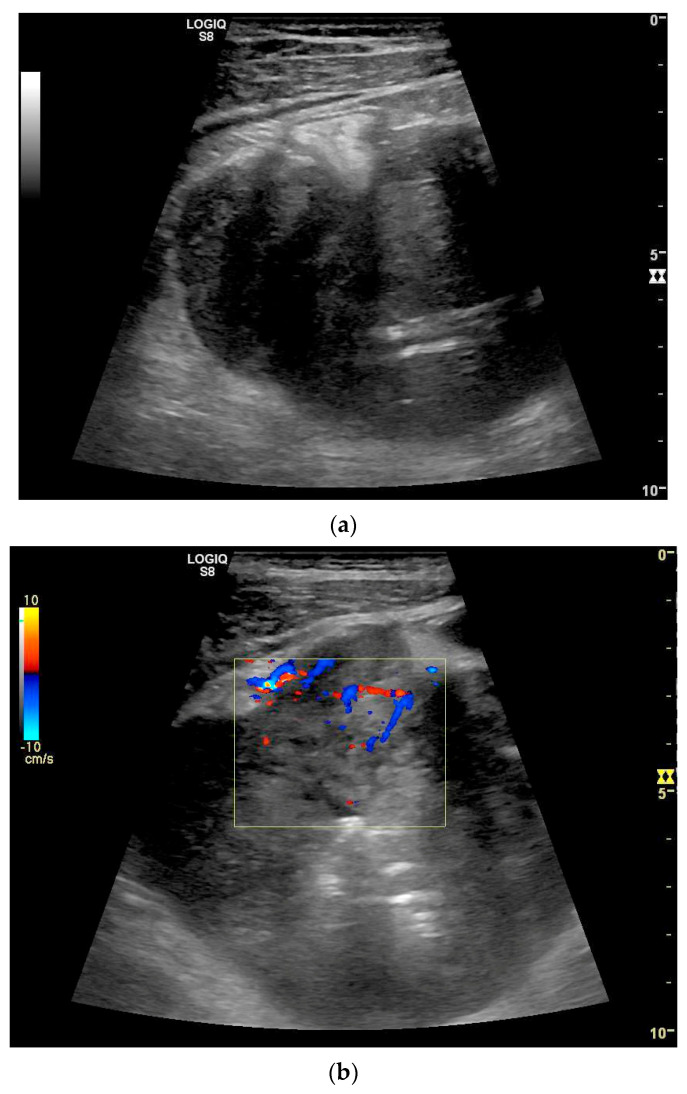
Desmoid fibromatosis of the terminal ileum. Initial pain in the right lower abdomen and suspicion of acute appendicitis. US revealed a polygonal tumor measuring up to 9 cm. B-mode US with reference to the small intestine (**a**), marginal vessels in duplex (**b**). CT revealed a tumor presumably in the terminal ileum with low contrast uptake, central necrosis, and air pockets. In addition, covered perforation was suspected. Surgical ileocecal resection was performed, followed by histological diagnosis of desmoid fibromatosis.

**Figure 15 healthcare-13-02776-f015:**
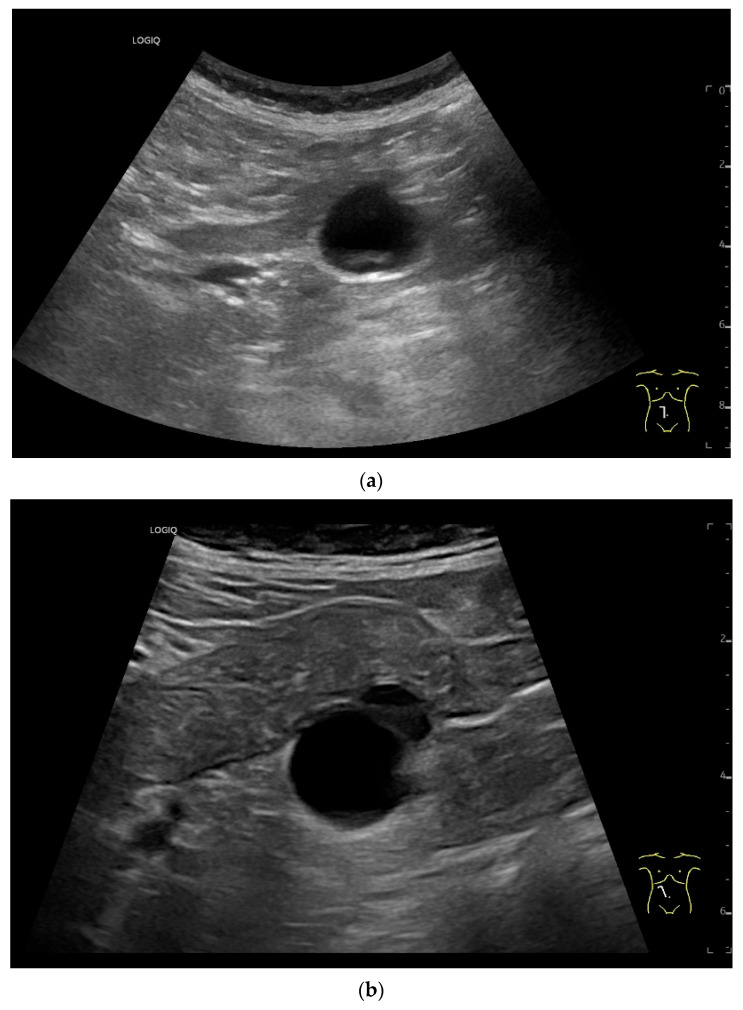
Cyst related to the serosa of the Jejunum. An incidental finding reveals a smoothly defined, non-echoic lesion in the right mid-abdomen using the abdominal sector transducer (**a**). Using the linear transducer, the cystic lesion can be visualized adjacent to the small intestine/jejunum (**b**). Kerckring folds are visible. The cystic lesion moves with the jejunum. The lesion is in contact with the small intestine wall (between the markers) (**c**). In CEUS with 2.0 mL SonoVue (linear transducer 9 MHz), the wall shows subtle enhancement. The lesion itself shows no enhancement and is therefore not solid, but liquid. The wall of the adjacent small intestine shows clear enhancement (**d**). Differential diagnoses include lymphangioma or mesothelial cyst. Since the elderly patient was completely symptom-free and showed no signs of gastrointestinal bleeding, no surgery was performed. The findings remained on follow-up.

**Figure 16 healthcare-13-02776-f016:**
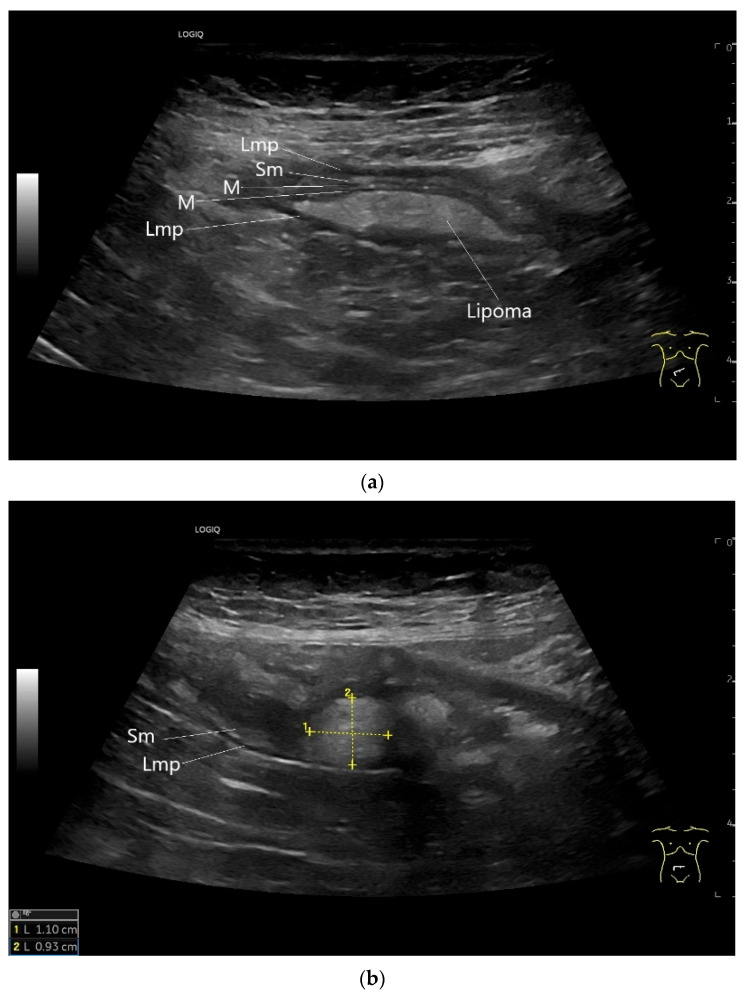
Lipoma in the ileum. An incidental finding reveals a homogeneous hyperechoic lesion in the right lower abdomen. This can be attributed to the ileum. The mass is shown in longitudinal section (**a**) and cross-section (**b**). The longitudinal section (**b**) shows the layer classification. M—mucosa; Sm—submucosa; Lmp—muscularis propria layer.

**Figure 17 healthcare-13-02776-f017:**
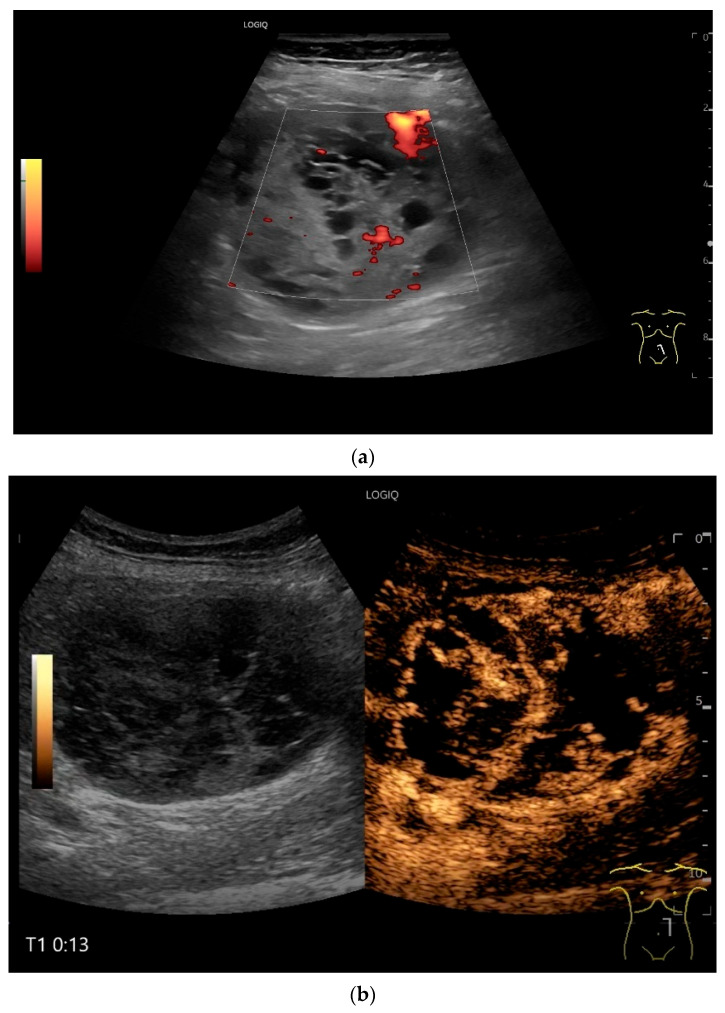
Schwannoma. Incidental finding of a 60 mm solid mass with nonechoic cystic components (**a**). In CEUS after 13 s (**b**) and 23 s (**c**), the echogenic components are increasingly heterogeneous and irregularly enhanced, thus corresponding to solid vital tissue. The mass was located in the left upper abdomen in the middle of the intestinal loops. US suggested a connection to the jejunum, while CT showed a tumor in the duodenojejunal flexure. Intraoperatively, a tumor was seen to the left of the mesenteric root. The first jejunal loop was adherent. In summary, a tumor was diagnosed, which corresponded to a schwannoma on histology.

**Figure 18 healthcare-13-02776-f018:**
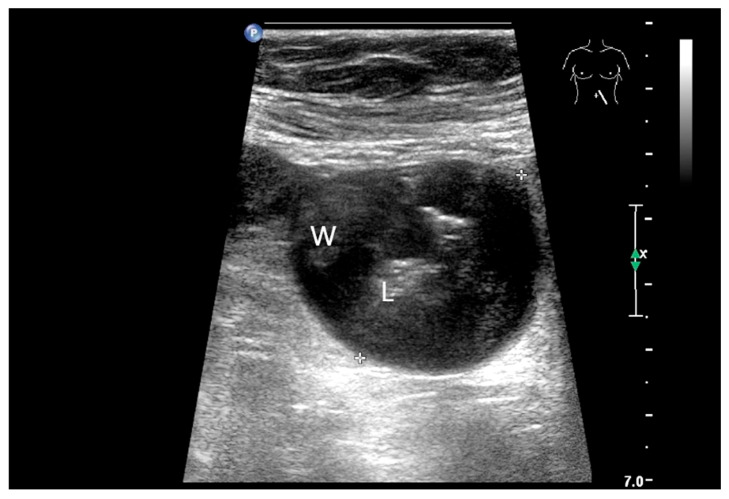
Small intestine metastasis of a malignant melanoma (between the markers). Significant hypoechoic wall thickening (W) with narrowed lumen reflex (L) and lumen obstruction.

**Figure 19 healthcare-13-02776-f019:**
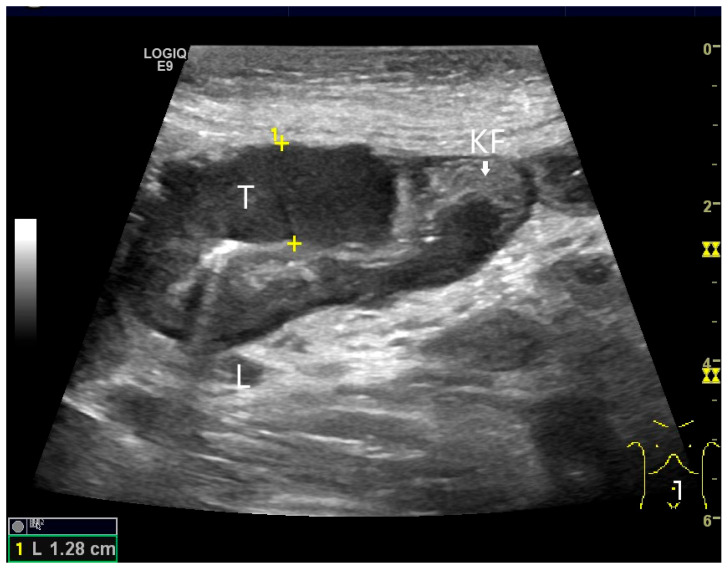
Small intestine metastasis of pleural mesothelioma. Hypoechoic tumorous wall thickening (T) between the markers. The normal wall with Kerckring’s folds (KF) is visible adjacent to it. Next to the small intestine is a round hypoechoic tumor-suspicious lymph node (L).

**Figure 20 healthcare-13-02776-f020:**
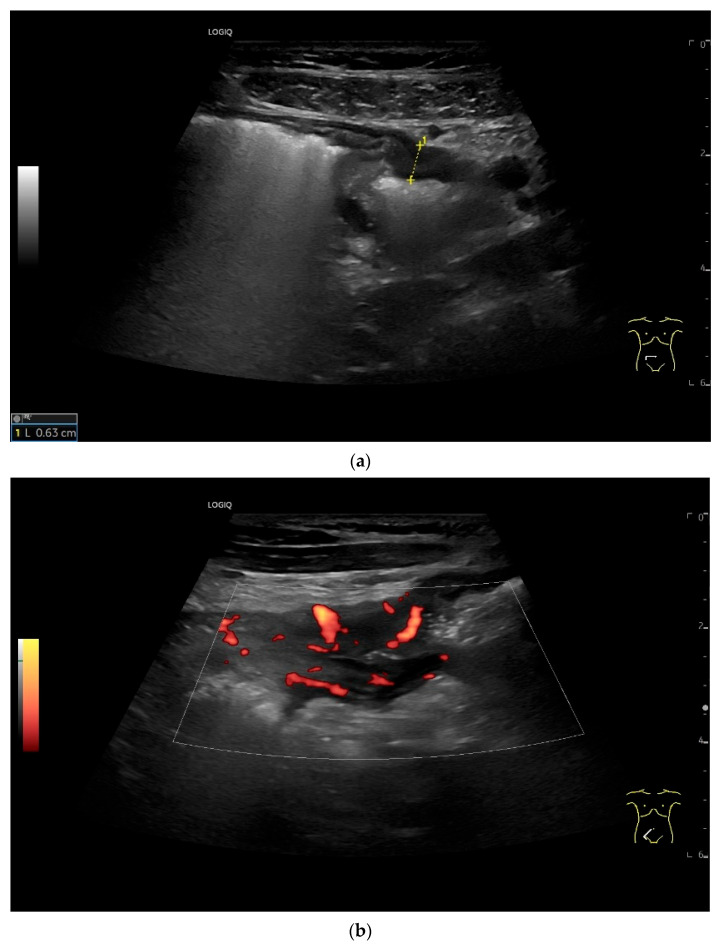
Crohn’s disease under anti-inflammatory therapy, anastomosis after ileocecal resection. Hypoechoic wall thickening of the neoterminal ileum without definable stratification (**a**). Numerous vascular reflexes are visible in color Doppler (**b**), indicating ongoing inflammation.

**Figure 21 healthcare-13-02776-f021:**
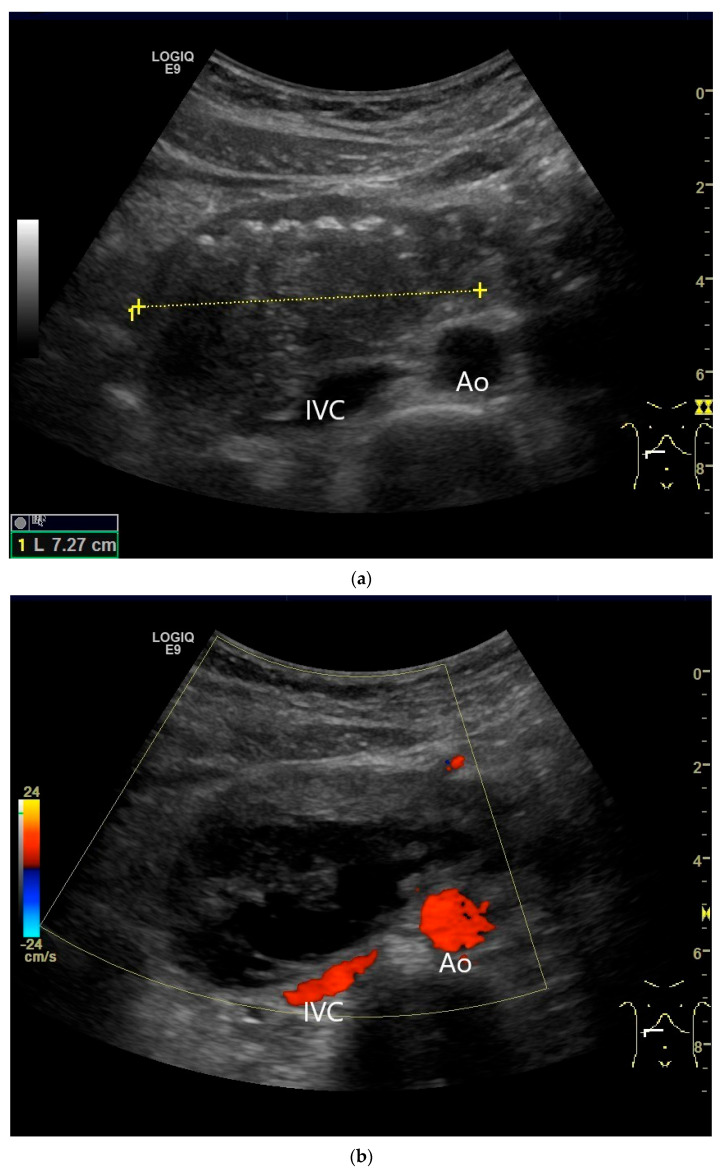
Duodenal wall hematoma after gastroscopy with forceps biopsy under dual antiplatelet therapy. Clinically, there was pain and lumen obstruction. Evident focal hypoechoic wall thickening (between the markers) with lumen obstruction (**a**), for which there was no correlation in the previous gastroscopy. No evidence of macro vessels in CDI (**b**). In CEUS, the mass is smoothly defined and completely non-enhanced (**c**). IVC—inferior caval vein, Ao—aorta.

**Figure 22 healthcare-13-02776-f022:**
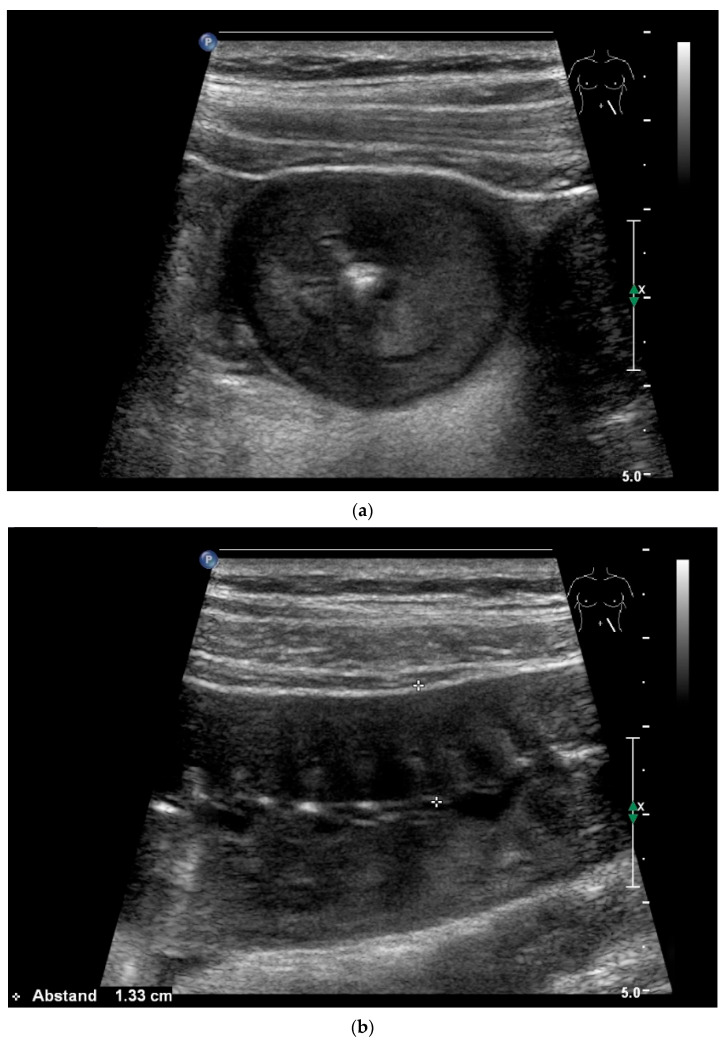
Jejunal wall hematoma with a Quick ratio <5%. Smoothly defined, distinct wall thickening and lumen obstruction. The wall stratification is indistinct, slightly hyperechoic, and clearly blurred and unfocused (**a**). The Kerckring folds are thickened and also very blurred (**b**).

**Figure 23 healthcare-13-02776-f023:**
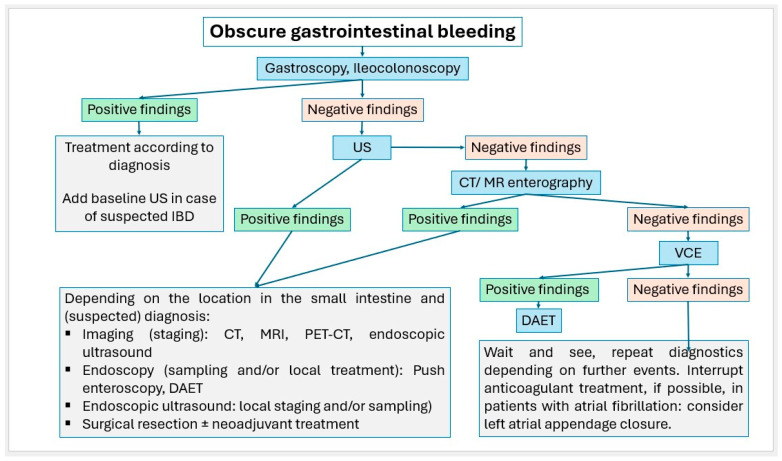
Algorithm for the stepwise approach to obscure gastrointestinal bleeding.

**Figure 24 healthcare-13-02776-f024:**
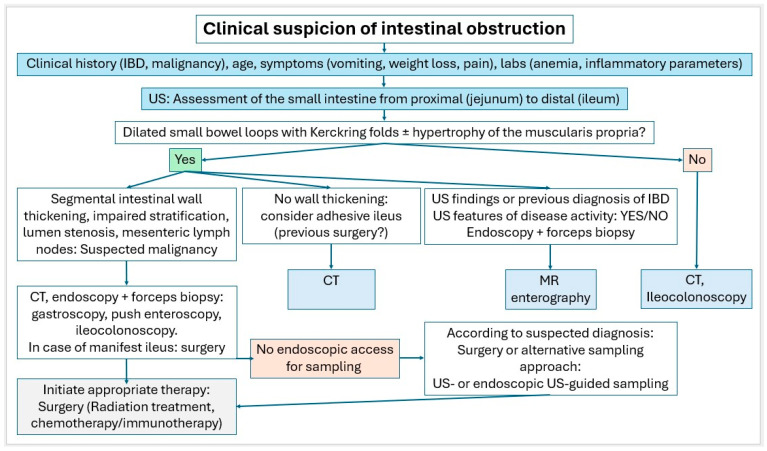
Algorithm in case of clinical suspicion of intestinal obstruction.

**Figure 25 healthcare-13-02776-f025:**
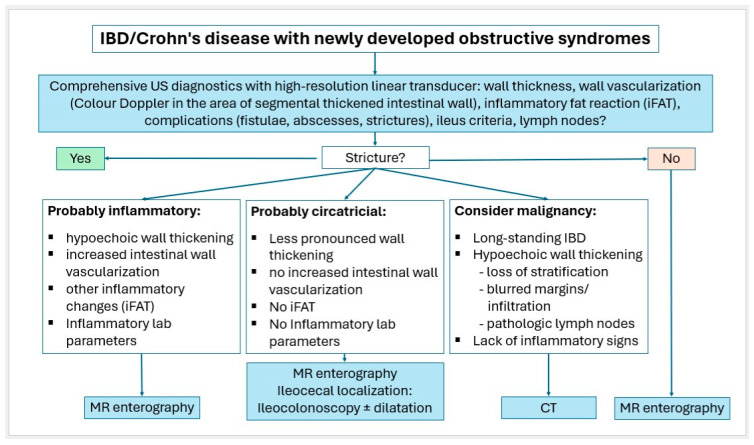
Algorithm for IBD/Crohn’s disease with newly developed obstructive syndromes.

**Table 1 healthcare-13-02776-t001:** Factors influencing the detection of small intestine tumors in US.

	Fukumoto et al. 2009 [41]	Fujita et al. 2015 [42]
**Number of small intestine tumors**	87	97
**Mean tumor size**	20.0 mm (range 1–150 mm)	N/A
**Transducer**	3.5 MHz and 7.0 MHz	3.5 MHz curved and 7.5 MHz linear array transducers
**Sensitivity of US**	26.4%	50.5%
**Specificity of US**	98.6%	100%
**PPV**	N/A	100%
**NPV**	N/A	90.9%
**Detection rate**		
All lesions	25%	52.6%
Lesions < 20 mm	1.8%	14.3%
Lesions > 20 mm	59.5%	91.7%
**Factors favoring detection**	- Size > 20 mm - Circumferential ulcerative lesions	- Size > 20 mm - (Mean size of detected lesions: 33.2 ± 16.3 mm) - Partial and circumferential ulcerative lesions - localization in the ileum (ileocecal) - malignancy (only 4.9% of malignant lesions were not detected)
**Factors favoring non-detection**	- Granular lateral spreading lesions (despite being large) - Location deep in the pelvic cavity - Hyperechoic lesions	- Size < 10 mm (mean size of non-detected lesions 8.7 ± 8.6 mm; 80.8% were benign) - Localization in the pelvic cavity, upper jejunum, or duodenum
**Detection according to shape**		
Granular lateral spreading	0% (0/24)	42.8% (3/7)
Flat elevated	33.3% (1/3)	N/A
Polypoid	50.0% (4/8)	22.0% (4/41)
Submucosal	16.6% (4/42)	47.1% (8/17))
Partial ulcerative	33.3% (2/6)	92.3% (12/13)
Circumferential ulcerative	100% (9/9)	100% (19/19)

**Table 2 healthcare-13-02776-t002:** Strengths and weaknesses of US compared to CT/MRI procedures, VCE, and DAET.

Method	Strengths	Weaknesses/Limitations	Sensitivity/ Specificity
**US**	Performable at any time, widely available, can be performed bedside. Dynamic and real-time examination allowing for detection of pathological peristalsis phenomena. Highest spatial and temporal resolution of all cross-sectional imaging modalities using high-frequency transducers, allowing delineation of the bowel wall layering and vascularization as well as assessment of the surrounding structures, Combination with CEUS to differentiate between nonvascularized and vascularized lesions and to delineate necrosis. Percutaneous US-guided sampling is possible if a histological diagnosis is required.	Detection depends on the size and shape of the small intestine tumor. Considerable examiner experience is required. Patient limitations include obesity, air overlap, and artifacts-causing from bowel contents. Deep-seated findings may be obscured and less visible. Systematic examination of the small bowel is time-consuming.	26.4%/98.6% [41] 53.1%/100%/PPV 100%/NPV 90.9% [42]
**CT**	Also a primary examination, readily available, not time-consuming. Allows excellent overview and multiplanar reconstruction. Can be used as an acute examination. Contrast enhancement allows assessment of vascularity	Restrictions due to elevated renal retention parameters. Radiation exposure. Limited spatial resolution. Limited soft-tissue contrast.	Sensitivity 87.7%/N/A [36]
**CT enterography**	High detection rate. Imaging of the intestinal wall and surrounding area.	Filling of the small intestine with contrast medium required. Limitations in elevated renal retention parameters. High radiation exposure.	75.9%/94.8% Accuracy 88.0% [38]
**MR enterography**	High detection rate. Imaging of the intestinal wall and surrounding area. No radiation exposure.	Filling of the small intestine with contrast medium required. Limited availability, time-consuming and relatively costly.	92.6%/99.0% Accuracy 96.7% [38]
**VCE**	Indicated in particular for the search for causative intestinal lesions in obscure bleeding	Bowel preparation/laxative measures, exclusion of stenosis by means of US or CT/CT-/MR enterography to reduce the risk of capsule retention. Subepithelial lesions can be overlooked, and external impressions can be misinterpreted. Rapid peristaltic waves can lead to forced capsule passage in the area of lesions. Bowel contents and blood impair the assessment.	Lesions in patients with suspicion on small intestine tumors: Compared to MRI 86%/98%, Accuracy 97% [43]
**DAET**	The only endoscopic modality which allows to examine the lumen and mucosa of the small intestine and to take forceps biopsies for histological evaluation of tumors	Not the entire small intestine is possible to be examined. Peroral (prograde) access or peranal (retrograde) access often have to be combined. Time-consuming, costly, limited availability. Laxative measures for bowel cleansing. High level of examination expertise required.	Based on surgery results, the accuracy rates of prograde and retrograde DAET for locating small intestinal tumors (adenocarcinoma, GIST and lymphoma) were 94.4%, 100% and 100%, respectively [28]

**Table 3 healthcare-13-02776-t003:** Lymphomas with manifestation in the small intestine according to WHO classification 2019 [2].

Lymphoma	Most Common Localization in the Gastrointestinal Tract	Appearance
**B-Cell-Lymphoma**		
Diffuse large B-cell lymphoma	Any part of the digestive tract, most commonly stomach and ileocecal region.	Often solitary, infiltrating tumor mass with ulceration of the mucosa. Mostly transmural infiltration.
Extranodal marginal zone lymphoma	At any site of the digestive system, most common stomach and small intestine.	Arising in mucosal or glandular tissues. Thickening of the mucosa. Mass-forming lesions occur.
Follicular lymphoma	Most often in the small intestine and colon.	Multiple nodular/polypoid lesions and elevated white patches
Duodenal-type follicular lymphoma	Most commonly in the second portion of the duodenum; commonly simultaneous involvement of the jejunum or ileum.	Located in the mucosa/submucosa and often result in a polypoid architecture. Nodular white submucosal deposits or white villus enlargement. Mostly without lymph node involvement.
Mantle cell lymphoma	Lymph nodes and any site of gastrointestinal tract.	Subclinical tumor cell infiltration of the gastrointestinal tract is possible, superficial ulcers, large tumor masses and diffuse thickening of the gastrointestinal mucosa. Presentation as multiple intestinal polyps (multiple lymphomatous polyposis).
Burkitt lymphoma	In the gastrointestinal tract the ileocecal region, stomach and small intestine are the most common.	Bulky disease and high tumor burden.
Plasmablastic lymphoma	Most commonly in the oral cavity of HIV-positive patients. The digestive tract is affected in 20% of cases, including the small intestine.	Thickening of the intestinal wall, multiple tumor nodules in the intestinal wall and mesentery of the small intestine and surrounding organs.
Posttransplant lymphoproliferative disorders (PTLD)	A spectrum of abnormal lymphatic hyperplasia to open neoplasia that occurs after organ transplantation. Often associated with Epstein-Barr virus (EBV) infection, but this is not mandatory. Most frequent in the distal jejunum and ileum.	Wall thickening and dilatation, eccentric mass, luminal ulceration, and short-segment intussusceptions
**T-Cell-Lymphoma**		
Enteropathy-associated T-cell lymphoma	Jejunum is most often involved, but other segments of the small intestine, and less commonly the stomach and colon, may be affected.	Often multifocal, with circumferential ulcers, ulcerated nodules, plaques and strictures. The mesentery and mesenteric lymph nodes may be involved.
Monomorphic epitheliotropic intestinal T-cell lymphoma (MEITL)	Mostly ulcerated masses in the small intestine or (more rarely) in the large intestine, duodenum or stomach. A small subset of patients may be present with multifocal gastrointestinal lesions. In addition to the mesenteric lymph nodes, manifestations in the liver, lungs and brain are possible.	Multiple elevated and/or ulcerated lesions.
Intestinal T-cell lymphoma NOS (ITCL)	The colon and the small intestine are the organs most frequently affected. Dissemination to regional lymph nodes and extra gastrointestinal sites is not uncommon.	Often an ulcerated, plaque-like appearance; sometimes protruding luminal masses.
Indolent T-cell lymphoproliferative disorder of the gastrointestinal tract	Mostly small intestine or colon. However, all sites of GI may be involved.	The affected mucosa is thickened, with conspicuous folds or nodules. In some tumors, the infiltrate gives rise to intestinal polyps that resemble lymphatic polyposis.
Extranodal NK/T-cell lymphoma, nasal-type	The gastrointestinal tract is the third most common site after the nasal cavity and the skin. Predominantly in the small and large intestine.	Gastrointestinal tract manifestations with tumor infiltration, mass formation, often accompanied by necrosis, ulceration, and/or perforation.

**Table 4 healthcare-13-02776-t004:** Mesenchymal tumors with manifestation in the small intestine according to WHO-classification 2019 [3].

Mesenchymal Tumor	Most Common Localization in the GIT	Appearance
**Benign**		
Inflammatory fibroid polyp	Most commonly in the stomach, followed by the ileum, but can develop throughout the GI tract.	Usually found in the submucosa but can also be localized in the mucosa. Ileal tumors are often larger.
Hemangioma	The small intestine is the most common location for hemangiomas and vascular malformations. Otherwise, they can occur anywhere.	Gastrointestinal tract hemangiomas grow either polypoid and intraluminal or are diffusely infiltrating submucosal lesions. They are purplish red to blue, soft and compressible.
Lipoma	Found in any part of the gastrointestinal tract. Rare in the small intestine and there predominantly located in the ileum.	Smoothly bordered oval/round lipomatous lesions, usually originating in the submucosa. Yellowish shimmering impressions on endoscopy, soft on contact.
Leiomyoma	Predominantly in the esophagus, colon and rectum and rare in the stomach and small intestine.	Smoothly bordered intramural tumors of different sizes.
Schwannoma	Mostly found in the stomach, very rare in other gastrointestinal areas including the small intestine.	Well-defined lesions originating in the submucosa and lamina muscularis propria. Solid and cystic parts.
Lymphangioma/lymphangiomatosis	Most commonly in the small intestine, followed by the colon and esophagus.	Either small (<2 cm) polypoid white or yellow mucosal lesions, usually discovered as incidentalomas, or larger masses with transmural spread or origin in the mesenterial adipose tissue.
PEComa/Angiomyolipoma	Rare tumors with occurrence in the colon, liver, small intestine, and pancreas.	Circumscribed unencapsulated masses may extend into the adjacent mesenteric tissue. Tumor size ranges from <1 cm to >20 cm.
Granular cell tumor	Most common in esophagus, colon and perianal region, very rare in the small intestine.	Small lesions of 0.1–3.0 cm, yellowish shimmering through.
Perineurinoma/benign fibroblastic polyp	May occur in the colon, only very rarely in the small intestine or stomach.	Small (0.2–0.6 cm) colonic polyps. More rarely non-polypoid lesions in the submucosa.
**Intermediate**		
Inflammatory myofibroblastic tumor	Most commonly located in the small intestine and colon.	Different sizes. The submucosa, the muscularis propria or the mesentery may be affected.
Desmoid fibromatosis	Most commonly in the mesentery of the small bowel, followed by the mesentery of the ileocolic region and the mesocolon.	Tends to be a large mass.
Solitary fibrous tumor	At any anatomical location including the small bowel mesentery.	Well circumscribed, unencapsulated, may become very large.
Kaposi sarcoma	It can manifest at any location within the gastrointestinal tract, most common in the stomach and duodenum.	Usually affects the mucosa at several localizations, reddish-blue or brown flat lesions, luminal polypoid nodules or ulcerating and hemorrhagic lesions on the mucosa, rarely as a transmural mass.
**Malignant**		
Gastrointestinal stromal tumor	Approximately 30% in the small intestine, 50–70% in the stomach.	Round, hypoechoic lesions with reference to the lamina muscularis propria.
Gastrointestinal clear cell sarcoma (CCS)/malignant gastrointestinal neuroectodermal tumor (GNET)	Most common locations are the small intestine, stomach, and colon.	Variable size, polypoid masses, often lobulated, often ulcerations
Leiomyosarcoma	Since GISTs have been classified, there have been only a few reports of leiomyosarcomas in the GI tract since 2000.	Polypoid intraluminal tumors or ulcerating, solid or cystic masses

**Table 5 healthcare-13-02776-t005:** Small intestine tumors and their appearance on US and differential diagnoses.

Small Intestinal Neoplasm	Appearance on US
**Local Findings**	
Adenocarcinoma	Hypoechoic wall thickening with infiltration into the surrounding mesentery depending on the stage. Loss of stratification depending on tumor stage. There is insufficient data to describe vascularization in CDI and CEUS. Focally enlarged (tumor) lymph nodes, distant (tumor) lymph nodes.
Neuroendocrine tumor	Small, nodular hypoechoic wall thickenings, mostly in the submucosa with spreading into the other layers. Usually with small vessels on CDI. Regionally enlarged lymph nodes. Multilocular manifestations are possible.
Lymphoma	Very pronounced wall thickening with marked hypoechogenicity. Large regional and distant lymph nodes. Look for splenic infiltration. Tumor vessels on CDI and hyperenhancement on CEUS. Heterogeneous hyperechogenicity of the mesentery with walling of the mesenteric vessels. Multiple localizations are possible.
GIST	Round hypoechoic masses, homogeneous or heterogeneous depending on size. They usually originate from the muscularis propria, which can be difficult to distinguish in US. Small vessels on CDI, hyperenhancement on CEUS. They move with the small intestine and can change position.
Lymphangioma	Anechoic cystic lesions related to the small intestine wall. No vessels on CDI, Nonenhancement on CEUS. They move with the small intestine.
Intestinal metastases	Round lesions or wall thickening. Take medical history into account. There is insufficient data to describe vascularization in CDI and CEUS. Assigning them to the small intestinal wall can be difficult. Round intestinal metastases must be differentiated from mesenchymal tumors and enlarged or tumorous lymph nodes.
**Indirect Signs**	
Intussusception	The small intestine proximal to the tumor is invaginated. More than five wall layers are seen in an onion-skin shape.
Proximal to the tumor	Dilated small bowel lumen, possibly hypertrophy of the muscle layer and hyperperistalsis of this bowel segment
**Differential Diagnoses**	
Crohn’s disease	Hypoechoic wall thickening, lumen obstruction, infiltration into surrounding tissue, lymph nodes. Very difficult or impossible to distinguish from inflammatory wall infiltration. Lack of response to anti-inflammatory/immunosuppressive therapy.
Tuberculosis	Asymmetric wall thickening with preserved but often blurred stratification. Intramural caseous necrosis/abscesses, inflammation extending beyond the wall. Enlarged hypoechoic mesenteric lymph nodes, ascites with fibrin strands, adhesions, hyperechoic thickening of the omentum and peritoneum; usually terminal ileum, ileocecal valve, and cecum. Patients from endemic areas, with immunosuppression/HIV.
Hematoma	Sudden severe pain, following abdominal trauma or anticoagulation therapy. Hypoechoic/nonechoic wall thickening. Lumen obstruction with dilatation of the proximal small intestine.

## Data Availability

The data presented in this study are available upon request from the corresponding author. Some of the data originates from articles of other authors (see references). The data are not publicly accessible, as the personal rights of the patients involved must be respected.

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
