# Peer review of "Small Intestine Tumors: Diagnostic Role of Multiparametric Ultrasound"

_healthcare, 2025, doi:10.3390/healthcare13212776_

Round 1
Reviewer 1 Report
Comments and Suggestions for Authors
Dear Authors,
Thank you for the opportunity to review your comprehensive review article on the diagnostic role of multiparametric ultrasound in small intestinal tumors. The topic is highly relevant, as small bowel tumors are rare, often detected late, and current diagnostic approaches rely heavily on CT, MRI, and invasive endoscopic techniques. Ultrasound remains underutilized in this setting, and your review draws attention to its potential clinical value.
Strengths
-
Comprehensive Literature Review – The manuscript systematically reviews adenocarcinomas, neuroendocrine tumors, lymphomas, and mesenchymal tumors, integrating epidemiology, clinical presentation, and imaging findings.
-
Methodological Transparency – The literature search strategy is clearly described, with PubMed-based keyword combinations and selection methods detailed (Figure 1).
-
Educational Value – Numerous ultrasound images (figures) provide readers with practical visual references that enhance the clinical teaching value.
-
Novel Perspective – Emphasizing multiparametric ultrasound (conventional, Doppler, CEUS, and US-guided biopsy) in a field dominated by cross-sectional imaging provides an original contribution.
-
Clinical Relevance – Highlighting detection thresholds (>20 mm), tumor-type specific appearances, and the value of CEUS in vascular characterization adds pragmatic insights for clinicians.
Areas for Improvement
-
Critical Appraisal – While the review is thorough, it is largely descriptive. Adding a critical comparison of ultrasound versus CT/MR modalities (sensitivity, specificity, accessibility, radiation, cost) in a structured table would strengthen the impact.
-
Novelty and Future Directions – The manuscript should more explicitly state what is new compared to prior reviews, and suggest research directions (e.g., AI-assisted ultrasound, multicenter prospective validation).
-
Balance of Content – Sections on lymphoma and mesenchymal tumors are highly detailed, whereas adenocarcinoma and early lesions receive comparatively less attention. A more balanced approach would be preferable.
-
Figures and Quality – Ensure that all figures are of publishable resolution, with consistent legends and annotations (arrows, markers). Some images appear cluttered and may need simplification.
-
Language – The English is generally clear but could benefit from light editing for conciseness and smoother flow. For example, long sentences could be shortened to improve readability.
-
Limitations – The limitations of ultrasound (operator dependency, patient habitus, gas interference) are acknowledged but should be emphasized more prominently, particularly for generalizability.
-
Recent Literature – The review includes many important references, but adding the most recent data from 2023–2024 (particularly on CEUS and AI applications) would enhance currency.
Minor Points
-
Standardize abbreviations early (e.g., NETs, GIST, CEUS).
-
In Table 1 and Table 2, simplify the text-heavy format to make them more reader-friendly.
-
A graphical abstract or summary figure comparing diagnostic modalities would improve accessibility.
Overall Evaluation
This is a well-structured and informative review that highlights the underrecognized role of multiparametric ultrasound in small intestinal tumors. With minor revisions focused on emphasizing novelty, balancing content, improving figure quality, and refining language, the manuscript will be suitable for publication.
Author Response
Reply to Reviewers
Dear Editors and Reviewers,
Thank you for the comments concerning our manuscript “Small Intestinal Tumors: Diagnostic Role of Multiparametric Ultrasound”, by Kathleen Möller, Christian Jenssen, Klaus Dirks, Alois Hollerweger, Heike Gottschall, Siegbert Faiss, Christoph F Dietrich.
The comments are important and helpful to improve our paper. Please find our detailed, point-by-point responses below. All the changes were marked in blue and/or highlighted yellow in our revised manuscript.
We hope that, following these thorough revisions, our paper will now meet the requirements for publication.
Sincerely yours,

Reviewer 2 Report
Comments and Suggestions for Authors
Recommendation: Accept, pending Minor Revisions
The manuscript entitled “Small Intestinal Tumors: Diagnostic Role of Multiparametric Ultrasound” is a well-structured and comprehensive review that addresses a clinically important but relatively underrepresented topic. The paper is competently written, supported by an extensive and up-to-date literature review, and enriched with high-quality figures, which significantly enhance its educational value. Therefore, I recommend its publication, pending minor revisions.
Abstract
-Consider adding a short sentence mentioning the limitations of ultrasound compared with other modalities.
Terminology
-There is variation between small bowel tumors and small intestinal tumors; it would be preferable to use one consistently.
Language
-Overall, the English is very good. A light copy-editing could further improve fluency and readability in some sentences.
Discussion and Reference
-It may be useful to describe the rare possibility of encountering multiple small intestinal tumors, sometimes in the context of multiple primary malignancies (MPMs). Highlighting this scenario would make the discussion more comprehensive, emphasizing that, although uncommon, this eventuality should be considered in the differential diagnosis. I would suggest two references that contribute both to defining the concept of MPMs and the role of diagnostic imaging in their detection. One of them, in particular, describes the unusual case of three small intestinal lesions identified simultaneously with a mucinous carcinoma of the cecum.
-Synchronous tumours detected during cancer patient staging: prevalence and patterns of occurrence in multidetector computed tomography. Pol J Radiol. 2020 May 26;85:e261-e270. doi: 10.5114/pjr.2020.95781. PMID: 32612725; PMCID: PMC7315052.
Comments on the Quality of English LanguageOverall, the English is very good. A light copy-editing could further improve fluency and readability in some sentences.
Author Response

(The authors gave the same response as above.)

Reviewer 3 Report
Comments and Suggestions for Authors
Dear Authors this is an interesting review with an atlas about the presentation of small intestinal tumors at bowel ultrasound. However, the review is too long and several changes are required.
Here my comments in details:
- remove the sentence at lines 39-40 with the relative ref.
- lines 89: what is GIT?
- line 99: add "the duodenum until the second prtion is examined"
- line 101: add "the remnant duodenum and upper jejunum"
- remove paragraphs 4.1, 4.2, 4.3, 4.4 and summarize these in a short chapter
- line 162: the thickness of the wall of small intestine is up to 3 mm; remove ref. 13
- remove line 226-229 and 231-233
- remove paragraph 4.7 and 4.8 if there are no specific results about the use of these techniques for SB tumors
- remove lines 286-290 and line 301 and 315-316
- remove lines 341-347
- line 322: add "in the ampullary region and in the appendix"
- remove lines 365-368, 370-372 and 375-376
- remove paragraphs from 8.1 to 8.8
- remove lines 621-624, 629-634 and 663-679
- figure 11 line 696: replace "this is related to the jejunum" with "this is located in the jejunum"
- remove paragraph 9.2 and 9.3 and add the details on intestinal ultrasound in the table 2
- linee 799: remove "delicate"
- remove lines 818-828 and 832-834
- remove lines 938-940
Author Response

(The authors gave the same response as above.)

Reviewer 4 Report
Comments and Suggestions for Authors
This narrative review addresses the diagnostic role of multiparametric transabdominal ultrasound (TUS/GIUS), CEUS, DCEUS, and related techniques in small intestinal tumors (adenomas, adenocarcinoma, NEN, lymphoma, GIST). The topic is clinically relevant and underrepresented. The manuscript is generally well organized, with useful figures and a broad literature sweep. However, the current version mixes narrative and systematic approaches without sufficient methodological rigor for a review claiming a defined search strategy, and key elements (diagnostic accuracy synthesis, standardization of technique, and a clear clinical pathway) need strengthening.
- Create summary tables (by tumor type and ≥/<20 mm) reporting sensitivity, specificity, PPV/NPV, accuracy, and key sonographic patterns (echotexture, stratification loss, vascularity). Consider forest plots if enough homogeneous data exist (even in an educational appendix).
-
Add a practical diagnostic algorithm (flowchart) for patients with:
-
obscure GI bleeding;
-
suspected SBO/partial obstruction;
-
suspected NEN;
-
IBD with new obstructive symptoms.
Include triggers for escalation from TUS → CEUS → CT/MR enterography → VCE → DA-enteroscopy/biopsy.
-
- Add practical criteria for when to attempt US-guided biopsy (size, location accessibility, vascular mapping with CDI/CEUS, avoidance of perforation), complication rates, and sample handling (core vs FNA) with examples for lymphoma vs GIST.
- Consider a forward-looking paragraph on ultrasound elastography, microvascular imaging (MV-Flow/SMI), and AI/deep learning for small-bowel lesion detection/characterization.
-
Use consistent terminology: choose “small bowel” or “small intestine” and apply uniformly. Standardize TUS/GIUS, DA-enteroscopy (DBE/SBE), VCE.
-
Replace stray “OP” with “OR” in the search string.
-
“Berlin, German” → Germany. Check all affiliations for grammar and punctuation consistency.
-
Fix spacing around punctuation (e.g., “Austria ;” → “Austria;”).
-
Normalize apostrophes (e.g., Vanek’s tumor, not “Vanek ́s”).
-
Ensure reference style uniformity; add complete details for [1–9] etc. Cross-check in-text percentages vs cited sources.
- Use SI spacing (e.g., 5–6 m; 12–20 mSv). Ensure decimals and percent symbols are consistently formatted (e.g., 52.8%, not 52.8 %).
Author Response

(The authors gave the same response as above.)
